# On the number of variables to use in principal component regression

**Ji Xu**
Columbia University
jixu@cs.columbia.edu

**Daniel Hsu**
Columbia University
djhsu@cs.columbia.edu

## Abstract

We study least squares linear regression over $N$ uncorrelated Gaussian features that are selected in order of decreasing variance. When the number of selected features $p$ is at most the sample size $n$, the estimator under consideration coincides with the principal component regression estimator; when $p > n$, the estimator is the least $\ell_2$ norm solution over the selected features. We give an average-case analysis of the out-of-sample prediction error as $p, n, N \to \infty$ with $p/N \to \alpha$ and $n/N \to \beta$, for some constants $\alpha \in [0, 1]$ and $\beta \in (0, 1)$. In this average-case setting, the prediction error exhibits a "double descent" shape as a function of $p$. We also establish conditions under which the minimum risk is achieved in the interpolating $(p > n)$ regime.

## 1   Introduction

In principal component regression (PCR), a linear model is fit to variables obtained using principal component analysis on the original covariates. Suppose the data consists of $n$ i.i.d. observations $(\boldsymbol{x}_1, y_1), \ldots, (\boldsymbol{x}_n, y_n)$ from $\mathbb{R}^N \times \mathbb{R}$. Let $\boldsymbol{X} := [\boldsymbol{x}_1 | \cdots | \boldsymbol{x}_n]^\top$ be the $n \times N$ design matrix, $\boldsymbol{y} := (y_1, \ldots, y_n)^\top$ be the $n$-dimensional vector of responses, and $\boldsymbol{\Sigma} := \mathbb{E}[\boldsymbol{x}_1 \boldsymbol{x}_1^\top] \in \mathbb{R}^{N \times N}$. Assuming $\boldsymbol{\Sigma}$ is known (as we do in this paper), the PCR fit is given by $\boldsymbol{V}(\boldsymbol{X}\boldsymbol{V})^+ \boldsymbol{y}$, where $\boldsymbol{V} \in \mathbb{R}^{N \times p}$ is the matrix of top $p$ (orthonormal) eigenvectors of $\boldsymbol{\Sigma}$, and $\boldsymbol{A}^+$ denotes the Moore-Penrose pseudo-inverse of $\boldsymbol{A}$. PCR notably addresses issues of multi-collinearity in under-determined $(n < N)$ settings, while avoiding saturation effects suffered by other regression methods such as ridge regression [1, 7, 12].

The critical parameter in PCR is the number of components $p$ to include in the regression. Nearly all previous analyses of variable selection have restricted attention to the $p < n$ regime [e.g., 4]. This restriction may seem benign, as conventional wisdom suggests that choosing $p > n$ leads to over-fitting. This paper aims to challenge this conventional wisdom in a particular setting for PCR.

We study the prediction error of the PCR fit for all values of $p$ in the under-determined regime. We assume the $\boldsymbol{x}_i$ are Gaussian and conduct an "average-case" analysis, where the "true" coefficient vector is randomly chosen from an isotropic prior distribution. Thus, all of the original variables in $\boldsymbol{x}_i$ are relevant but weak in terms of predicting the response. When the eigenvalues of $\boldsymbol{\Sigma}$ exhibit some decay, one expects diminishing returns as $p$ increases. It is often suggested to find a value of $p$ that balances bias and variance, and such a value of $p$ can be found in the $p < n$ regime.

However, we show that when $p > n$, the prediction error can again be decreasing with $p$. This phenomenon—the second descent of the so-called "double descent" risk curve [2]—has been observed in a number of scenarios and for many different machine learning models (where $p$ is regarded as a nominal number of model parameters) [2, 3, 8, 13, 17]. In these previous studies, the limiting risk as $p \to \infty$ was often (but not always) observed to be lower than the best risk achieved in the $p < n$ regime. We prove that this phenomenon occurs with PCR in our data model: the lowest prediction error is achieved at some $p > n$, rather than any $p < n$.

**Our data model.** Our data $(\boldsymbol{x}_1, y_1), \ldots, (\boldsymbol{x}_n, y_n)$ are assumed to be i.i.d. with $\boldsymbol{x}_i \sim \mathcal{N}(\mathbf{0}, \boldsymbol{\Sigma})$, and

$$y_i = \boldsymbol{x}_i^\top \boldsymbol{\theta} + w_i.$$

Here, $w_1, \ldots, w_n$ are i.i.d. $\mathcal{N}(0, \sigma^2)$ noise variables, and $\boldsymbol{\theta} \in \mathbb{R}^N$ is the true coefficient vector. We assume, without loss of generality, that $\boldsymbol{\Sigma}$ is diagonal. In fact, we shall take $\boldsymbol{\Sigma} := \mathrm{diag}(\lambda_1, \ldots, \lambda_N)$ with distinct positive eigenvalues $\lambda_1 > \cdots > \lambda_N > 0$. The prediction (squared) error of $\boldsymbol{\theta}' \in \mathbb{R}^N$ is $\mathbb{E}_{\boldsymbol{x},y}[(y - \boldsymbol{x}^\top \boldsymbol{\theta}')^2]$, where $(\boldsymbol{x}, y)$ is an independent copy of $(\boldsymbol{x}_1, y_1)$.

*Some notation.* For a vector $\boldsymbol{v} \in \mathbb{R}^N$, let $\boldsymbol{v}_P \in \mathbb{R}^p$ denote the sub-vector of the first $p$ entries of $\boldsymbol{v}$, and let $\boldsymbol{v}_{P^c} \in \mathbb{R}^{N-p}$ denote the sub-vector of the last $N - p$ entries. Similarly, for a matrix $\boldsymbol{M} \in \mathbb{R}^{n \times N}$, let $\boldsymbol{M}_P \in \mathbb{R}^{n \times p}$ denote the sub-matrix of the first $p$ columns of $\boldsymbol{M}$, and let $\boldsymbol{M}_{P^c} \in \mathbb{R}^{n \times (N-p)}$ denote the sub-matrix of the last $N - p$ columns.

Recall that PCR selects components in order of decreasing $\lambda_j$. So, using the notation from above, the PCR estimator $\hat{\boldsymbol{\theta}}$ for $\boldsymbol{\theta}$ is defined by

$$\hat{\boldsymbol{\theta}}_P := \begin{cases} (\boldsymbol{X}_P^\top \boldsymbol{X}_P)^{-1} \boldsymbol{X}_P^\top \boldsymbol{y} & \text{if } p \le n, \\ \boldsymbol{X}_P^\top (\boldsymbol{X}_P \boldsymbol{X}_P^\top)^{-1} \boldsymbol{y} & \text{if } p > n; \end{cases} \qquad \hat{\boldsymbol{\theta}}_{P^c} := \mathbf{0}. \tag{1}$$

(Recall that $\boldsymbol{X} := [\boldsymbol{x}_1 | \cdots | \boldsymbol{x}_n]^\top$ and $\boldsymbol{y} := (y_1, \ldots, y_n)^\top$; also, the matrices being inverted above are, indeed, invertible with probability 1.) The prediction error of the PCR estimate $\hat{\boldsymbol{\theta}}$ is denoted by

$$\mathrm{Error} := \mathbb{E}_{\boldsymbol{x},y}[(y - \boldsymbol{x}^\top \hat{\boldsymbol{\theta}})^2].$$

Observe that the (squared) correlation between the response and the $j$th variable is proportional to $\lambda_j \theta_j^2$, but PCR selects variables only on the basis of the $\lambda_j$. So, for a worst-case $\boldsymbol{\theta}$, PCR may be unlucky and end up selecting the $p$ least correlated variables. To avoid this worst-case scenario, we consider an "average-case" analysis, where the true coefficient vector $\boldsymbol{\theta}$ is independently drawn from an isotropic prior distribution:

$$\mathbb{E}_{\boldsymbol{\theta}}[\boldsymbol{\theta}] = \mathbf{0}, \quad \mathbb{E}_{\boldsymbol{\theta}}[\boldsymbol{\theta} \boldsymbol{\theta}^\top] = \boldsymbol{I}. \tag{2}$$

We will study the random quantity $\mathbb{E}_{\boldsymbol{w},\boldsymbol{\theta}}[\mathrm{Error}]$, where the expectation is conditional on the design matrix $\boldsymbol{X}$, but averages over the observation noise $\boldsymbol{w} = (w_1, \ldots, w_n)$ and random choice of $\boldsymbol{\theta}$.

Our analysis uses high-dimensional asymptotic considerations to study the under-determined $(n < N)$ regression problem, letting $p, n, N \to \infty$ with $p/N \to \alpha$ and $n/N \to \beta$ for some fixed constants $\alpha \in [0, 1]$ and $\beta \in (0, 1)$. We are primarily interested in the limiting value of $\mathbb{E}_{\boldsymbol{w},\boldsymbol{\theta}}[\mathrm{Error}]$, which is the *asymptotic risk*.

**Our results.** In Section 2, we give an exact expression for the asymptotic risk in the case where the eigenvalues of $\boldsymbol{\Sigma}$ exhibit polynomial decay, namely $\lambda_j = j^{-\kappa}$ for a fixed constant $\kappa > 0$. Our expression covers both the $p < n$ and $p > n$ regimes, and we find that the smallest asymptotic risk can be achieved with $p > n$ (or equivalently, $\alpha > \beta$) in noiseless settings. In noisy settings, the comparison of the $p < n$ and $p > n$ regimes depends crucially on the exponent $\kappa$.

In Section 3, we relax the condition on the eigenvalues, and instead just assume that the empirical distribution of the $c_N \lambda_j$, for some suitable sequence $(c_N)_{N \ge 1}$, has a "nice" limiting distribution. We obtain results similar to those in Section 2 using a slightly different variable selection rule.

Our analyses permit a $1 - o(1)$ fraction of $\lambda_j$'s to converge to zero as $p, n, N \to \infty$. (In particular, the $c_N$ may go to infinity.) This makes our analysis technically non-trivial and more generally applicable.

The proofs of the results are detailed in the full version of the paper [19].

**Related works.** Strategies for choosing the optimal value of $p$ in PCR (e.g., cross validation, variance inflation factors) are typically only studied in the $p < n$ regime [9]. For instance, the exact risk of PCR as a function of $p$ for Gaussian designs can be extracted from the analysis of Breiman and Freedman [4], but only for the $p < n$ regime.

The high-dimensional analyses of ridge regression by Dicker [5], Dobriban and Wager [6], Hastie et al. [8] are closely related to our work. Indeed, for fixed $p$, the PCR estimator (or "ridgeless" estimator) is obtained by taking the ridge regularization parameter to zero. These analyses extend

beyond the Gaussian design setting that we consider, but are restricted to cases where either all eigenvalues of $\Sigma$ remain bounded below by an absolute constant as $N \to \infty$, or where the ridge regularization parameter is held at some positive constant.

The "double descent" phenomenon was observed by several researchers [e.g., 2, 8, 13, 17] for a variety of machine learning models such as neural networks and ensemble methods. Belkin et al. [3], Hastie et al. [8], Muthukumar et al. [13] provide statistical explanations for this phenomenon by studying the behavior of the minimum $\ell_2$ norm linear fit with $p > n$. The analysis of Muthukumar et al. [13] restricts attention to correctly-specified linear models (i.e., $p = N$ in our notation) and shows some potential benefits of the $p > n$ regime. A related analysis of estimation variance was carried out by Neal et al. [14]. The analysis of Belkin et al. [3] studies an isotropic Gaussian design that is otherwise similar to our setup, as well as a Fourier design that is related to the random Fourier features of Rahimi and Recht [15]. The analyses of Hastie et al. [8] look at more general and non-isotropic designs (and, in fact, certain non-linear models related to neural networks!), but as mentioned before, they assume the eigenvalues of $\Sigma$ are bounded away from zero. While their "misspecified" setting appears to be similar to our setup, we note that varying their $p/n$ parameter (which they call $\gamma$) changes the statistical problem under consideration. In contrast, our analysis looks at the effect of choosing different $p$ on the same statistical problem, and thus is able to shed light on the number of variables one should use in principal component regression.

**Notations for asymptotics.**  For any two random quantities $X$ and $Y$, we use the notation $X \xrightarrow{\mathrm{P}} Y$ to mean that $X = Y + o_{\mathrm{p}}(Y)$ as $n, p, N \to \infty$. Similarly, for any two non-random quantities $X$ and $Y$, we use the notation $X \to Y$ to mean that $X = Y + o(Y)$ as $n, p, N \to \infty$. Finally, we say that $X > Y$ holds in probability if $\Pr(X > Y) \to 1$ as $n, p, N \to \infty$.

## 2  Analysis under polynomial eigenvalue decay

In this section, we analyze the asymptotic risk of PCR under the following assumptions:

**A.1** There exists a constant $\kappa > 0$ such that $\lambda_j = j^{-\kappa}$ for all $j = 1, \dots, N$.

**A.2** There exist constants $\alpha \in [0, 1]$ and $\beta \in (0, 1)$ such that $p/N \to \alpha$ and $n/N \to \beta$ as $p, n, N \to \infty$.

Assumption **A.1** implies that the eigenvalues of $\Sigma$ decay to zero at a polynomial rate, while Assumption **A.2** is a standard scaling for high-dimensional asymptotic analysis.

We also assume in this section that there is no observation noise, i.e., $\mathrm{var}(w_i) = \sigma^2 = 0$. In the noiseless setting, the asymptotic risk is the limiting value of $\mathbb{E}_{\boldsymbol{\theta}}[\mathrm{Error}]$. Results for the noisy setting are stated in Appendix C.

### 2.1  Main results

Our first theorem provides characterizes the asymptotic risk when $\alpha < \beta$. Define the functions $h_\kappa$ and $\mathcal{R}_\kappa$ on $(0, \beta)$:

$$h_\kappa(\alpha) \; := \; \frac{\beta}{\alpha} - \int_\alpha^1 t^{\kappa-2}\, \mathrm{d}t - 1, \quad \text{for all } \alpha < \beta; \tag{3}$$

$$\mathcal{R}_\kappa(\alpha) \; := \; N^{1-\kappa} \int_\alpha^1 t^{-\kappa}\, \mathrm{d}t \cdot \frac{\beta}{\beta - \alpha}, \quad \text{for all } \alpha < \beta. \tag{4}$$

**Theorem 1.** *Assume **A.1** with constant $\kappa$; **A.2** with constants $\alpha$ and $\beta$; $\sigma^2 = 0$; and $\alpha < \beta$. Then*

$$\mathbb{E}_{\boldsymbol{\theta}}[\mathrm{Error}] \; \xrightarrow{\mathrm{P}} \; \mathcal{R}_\kappa(\alpha).$$

*Furthermore, the equation $h_\kappa(\alpha) = 0$ has a unique solution $\alpha^*$ over the interval $(0, \beta)$, and $\mathcal{R}_\kappa(\alpha)$ is decreasing on $\alpha \in [0, \alpha^*)$ and increasing on $\alpha \in (\alpha^*, \beta)$. Finally,*

$$\mathcal{R}_\kappa(\alpha^*) \; = \; \min_{0 \leq \alpha < \beta} \mathcal{R}_\kappa(\alpha) \; = \; N^{1-\kappa} \frac{\beta}{(\alpha^*)^\kappa}. \tag{5}$$

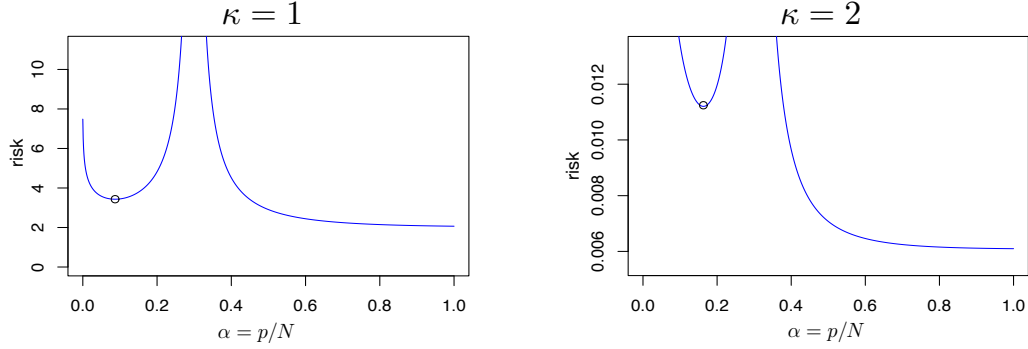

Figure 1: The asymptotic risk function $\mathcal{R}_\kappa$ as a function of $\alpha$ (with $n = 300$, $N = 1000$, $\beta = n/N = 0.3$ and $\kappa = 1, 2$ respectively). The location of $\alpha^*$ from Theorem 1 is marked with a black circle. In both cases, the asymptotic risk at $\alpha = 1$ is lower than the asymptotic risk at $\alpha^*$.

The proof of Theorem 1 is sketched in Section 2.2, with some details left to Appendix A. Theorem 1 supports the well-known intuition that the risk curve is "U-shaped" in the $p < n$ regime. Our next theorem, however, shows a very different behavior when $\alpha > \beta$.

Formally define $m_\kappa(z)$ for $z \leq 0$ to be the smallest positive solution to the equation

$$- z \;=\; \frac{1}{m_\kappa(z)} - \frac{1}{\beta} \int_{\alpha^{-\kappa}}^{\infty} \frac{1}{\kappa t^{1/\kappa}(1 + t \cdot m_\kappa(z))}\, \mathrm{d}t, \tag{6}$$

and let $m'_\kappa(\cdot)$ denote the derivative of $m_\kappa(\cdot)$. Also define the function $\mathcal{R}_\kappa$ on $(\beta, 1]$:

$$\mathcal{R}_\kappa(\alpha) \;:=\; N^{1-\kappa}\left( \frac{\beta}{m_\kappa(0)} + \int_\alpha^1 t^{-\kappa}\, \mathrm{d}t \cdot \frac{m'_\kappa(0)}{m_\kappa(0)^2} \right), \quad \text{for all } \alpha > \beta. \tag{7}$$

**Theorem 2.** *Assume A.1 with constant $\kappa$; A.2 with constants $\alpha$ and $\beta$; $\sigma^2 = 0$; and $\alpha > \beta$. The function $m_\kappa$ and its derivative $m'_\kappa$ are well-defined and positive at $z = 0$ (and hence $\mathcal{R}_\kappa(\alpha)$ is well-defined for all $\alpha > \beta$). Moreover,*

$$\mathbb{E}_{\boldsymbol{\theta}}[\text{Error}] \;\overset{\mathrm{p}}{\to}\; \mathcal{R}_\kappa(\alpha).$$

The proof of Theorem 2 is sketched in Section 2.3, with some details left to Appendix B.

We plot the asymptotic risk function $\mathcal{R}_\kappa$ in Figure 1 for two different values of $\kappa$, both with $\beta = 0.3$. (In simulations, we find that $\mathbb{E}_{\boldsymbol{\theta}}[\text{Error}]$ matches these curves very closely for sample sizes as small as $n = 300$.) For both values of $\kappa \in \{1, 2\}$, we observe the striking "double descent" behavior as found in previous studies [e.g., 2]. Moreover, we see that the asymptotic risk at $\alpha = 1$ is smaller than the minimum asymptotic risk achieved at any $\alpha < \beta$. This, in fact, happens for all values of $\kappa > 0$, as we claim in the next theorem.

**Theorem 3.** *Assume A.1 with constant $\kappa$, A.2 with constants $\alpha$ and $\beta$, $\sigma^2 = 0$. Let $\alpha^*$ be the minimizer of $\mathcal{R}_\kappa$ over the interval $[0, \beta)$. Then $\limsup_N \mathcal{R}_\kappa(1)/\mathcal{R}_\kappa(\alpha^*) < 1$. Moreover, $\mathcal{R}_\kappa(\alpha)/\mathcal{R}_\kappa(1) \to \infty$ as $\alpha \to \beta^-$.*

The proof of Theorem 3 is given in Section 2.4. Theorem 3 shows that the asymptotic risk exhibits a second decrease somewhere in the $p > n$ regime when $N$ is sufficiently large, and moreover, that it is possible to find a value of $p$ in this $p > n$ regime to achieve a lower asymptotic risk than any $p < n$.

In the noisy setting (see Appendix C), it is possible for the asymptotic risk to be dominated by the noise, in which case the minimum asymptotic risk is in fact achieved by $\alpha = 0$ (i.e., $p = o(n)$). However, there exists a regime with $\sigma^2 > 0$ in which we have the same conclusion as in Theorem 3.

## 2.2 Proof sketch for Theorem 1

We first show that $h_\kappa(\alpha) = 0$ has a unique solution on $(0, \beta)$. Define $\tilde{h}_\kappa(\alpha) := \alpha^{1-\kappa} h_\kappa(\alpha)$. We shall show that $\tilde{h}_\kappa(\alpha) = 0$ has a unique solution on $(0, \beta)$, which in turn immediately implies that $h_\kappa(\alpha) = 0$ also has a unique solution on the same interval. Observe that

$$\frac{\mathrm{d}\tilde{h}_\kappa(\alpha)}{\mathrm{d}\alpha} = \frac{-\kappa\beta + \kappa\alpha}{\alpha^{1+\kappa}} < 0. \tag{8}$$

Hence, the function $\tilde{h}_\kappa(\alpha)$ is strictly decreasing on $\alpha \in (0, \beta]$. Furthermore, we have

$$\tilde{h}_\kappa(\alpha) > 0 \text{ as } \alpha \to 0^+, \quad \text{and} \quad \tilde{h}_\kappa(\alpha) < 0 \text{ at } \alpha = \beta. \tag{9}$$

Because $\tilde{h}_\kappa$ is continuous, it follows that the equation $\tilde{h}_\kappa(\alpha) = 0$ has a unique solution on $(0, \beta)$.

We now prove $\mathbb{E}_{\boldsymbol{\theta}}[\text{Error}] \xrightarrow{\text{P}} \mathcal{R}_\kappa(\alpha)$. Since the proof only requires standard techniques, we just sketch the main ideas in this section, and leave the full proof to Appendix A. First, since $\alpha < \beta$, for large enough $N$, we have $p < n$. Then the prediction error is given by

$$\text{Error} = \mathbb{E}_{\boldsymbol{x},y}[(y - \boldsymbol{x}^\top \hat{\boldsymbol{\theta}})^2] = \|\boldsymbol{\Sigma}_P^{1/2} \left(\boldsymbol{X}_P^\top \boldsymbol{X}_P\right)^{-1} \boldsymbol{X}_P^\top \boldsymbol{X}_{P^c} \boldsymbol{\theta}_{P^c}\|^2 + \|\boldsymbol{\Sigma}_{P^c}^{1/2} \boldsymbol{\theta}_{P^c}\|^2,$$

where $\boldsymbol{\Sigma}_P \in \mathbb{R}^{p \times p}$ and $\boldsymbol{\Sigma}_{P^c} \in \mathbb{R}^{(N-p) \times (N-p)}$ are two diagonal matrices whose diagonal elements are the first $p$ and last $N - p$ diagonal elements of $\boldsymbol{\Sigma}$, respectively. By (2), we have

$$\mathbb{E}_{\boldsymbol{\theta}}[\text{Error}] = \text{tr}(\boldsymbol{X}_{P^c}^\top \boldsymbol{X}_P \left(\boldsymbol{X}_P^\top \boldsymbol{X}_P\right)^{-1} \boldsymbol{\Sigma}_P \left(\boldsymbol{X}_P^\top \boldsymbol{X}_P\right)^{-1} \boldsymbol{X}_P^\top \boldsymbol{X}_{P^c}) + \text{tr}(\boldsymbol{\Sigma}_{P^c}).$$

Note that $\boldsymbol{X}_{P^c}$ is independent of $\boldsymbol{X}_P$, thus, given $\boldsymbol{X}_P$, the trace that includes $\boldsymbol{X}_{P^c}$ is a sum of $N - p$ independent random variables. Therefore, we have

$$\begin{aligned} \mathbb{E}_{\boldsymbol{\theta}}[\text{Error}] &\xrightarrow{\text{P}} \text{tr}(\boldsymbol{\Sigma}_{P^c}) \cdot (\text{tr}((\boldsymbol{X}_P^\top \boldsymbol{X}_P)^{-1} \boldsymbol{\Sigma}_P) + 1) \\ &= \text{tr}(\boldsymbol{\Sigma}_{P^c}) \cdot (\text{tr}((\bar{\boldsymbol{X}}_P^\top \bar{\boldsymbol{X}}_P)^{-1}) + 1) \\ &\xrightarrow{\text{P}} \text{tr}(\boldsymbol{\Sigma}_{P^c}) \frac{\beta}{\beta - \alpha}, \end{aligned}$$

where $\bar{\boldsymbol{X}}_P := \boldsymbol{X}_P \boldsymbol{\Sigma}_P^{-1/2}$ is a standard Gaussian matrix. The first line above uses Markov's inequality to show that $\mathbb{E}_{\boldsymbol{\theta}}[\text{Error}]$ converges in probability to $\mathbb{E}_{\boldsymbol{\theta}, \boldsymbol{X}_{P^c}}[\text{Error}]$. The third line above uses Assumption **A.2** and the fact that $\bar{\boldsymbol{X}}_P^\top \bar{\boldsymbol{X}}_P$ is a standard Wishart matrix $\mathcal{W}_p(\boldsymbol{I}, n)$. So, to prove (4), we just need to compute $\text{tr}(\boldsymbol{\Sigma}_{P^c})$. Note that $\int_s^{s+1} t^{-\kappa} \, \mathrm{d}t < s^{-\kappa} < \int_{s-1}^s t^{-\kappa} \, \mathrm{d}t$. Hence, we have

$$\int_{p+1}^N \frac{N^\kappa}{t^\kappa} \, \mathrm{d}t \cdot \frac{1}{N} < \frac{N^\kappa}{N} \sum_{i=p+1}^N \frac{1}{i^\kappa} = N^{\kappa-1} \text{tr}(\boldsymbol{\Sigma}_{P^c}) < \int_p^N \frac{N^\kappa}{t^\kappa} \, \mathrm{d}t \cdot \frac{1}{N}. \tag{10}$$

Therefore, we have $\text{tr}(\boldsymbol{\Sigma}_{P^c}) \to N^{1-\kappa} \int_\alpha^1 t^{-\kappa} \, \mathrm{d}t$ as $p \to \infty$, and thus we have $\mathbb{E}_{\boldsymbol{\theta}}[\text{Error}] \xrightarrow{\text{P}} \mathcal{R}_\kappa(\alpha)$.

Finally, to prove (5), we analyze the shape of $\mathcal{R}_\kappa(\alpha)$ to find its minimum value over $\alpha < \beta$. We take the derivative of $g_\kappa(\alpha) := N^{\kappa-1} \mathcal{R}_\kappa(\alpha)$:

$$\frac{\mathrm{d}g_\kappa(\alpha)}{\mathrm{d}\alpha} = \beta \cdot \frac{\alpha^{1-\kappa} - \beta\alpha^{-\kappa} + \int_\alpha^1 t^{-\kappa} \, \mathrm{d}t}{(\beta - \alpha)^2} = \frac{-\beta\alpha^{1-\kappa} \cdot h_\kappa(\alpha)}{(\beta - \alpha)^2}. \tag{11}$$

Using (8) and (9), we deduce that $\mathcal{R}_\kappa(\alpha)$ first decreases and then increases as a function of $\alpha$ in the interval $(0, \beta)$. Therefore, the minimum risk is achieved at the unique solution $\alpha^*$ of the equation $h_\kappa(\alpha) = 0$ over the interval $(0, \beta)$. Equation (11) also implies $\int_{\alpha^*}^1 t^{-\kappa} \, \mathrm{d}t = (\beta - \alpha^*)(\alpha^*)^{-\kappa}$. Hence, the minimum risk is given by

$$\min_{\alpha < \beta} \mathcal{R}_\kappa(\alpha) = N^{1-\kappa} \frac{\beta}{\beta - \alpha^*} \int_{\alpha^*}^1 t^{-\kappa} \, \mathrm{d}t = N^{1-\kappa} \frac{\beta}{(\alpha^*)^\kappa}.$$

## 2.3 Proof sketch for Theorem 2

We first show that $m_\kappa(0)$ is well-defined. Consider the RHS expression from Equation (6) evaluated at $z = 0$; by a change-of-variable in the integral, we have

$$\frac{1}{m} - \frac{1}{\beta} \int_{\alpha^{-\kappa}}^\infty \frac{\kappa^{-1} t^{-1/\kappa}}{1 + t \cdot m} \, \mathrm{d}t = \frac{1}{\beta \alpha m^{1 - 1/\kappa}} \left( \frac{\beta}{m^{1/\kappa}/\alpha} - \alpha \int_{m^{1/\kappa}/\alpha}^\infty \frac{t^{\kappa - 2}}{1 + t^\kappa} \, \mathrm{d}t \right), \quad (12)$$

where $m = m_\kappa(0)$. So, we just need to show that $q_\kappa(s, \alpha) = 0$ has a unique solution $s_\kappa^*$ for $s$ over the positive real line, where $q_\kappa(s, \alpha)$ is defined by

$$q_\kappa(s, \alpha) := \frac{\beta}{s} - \alpha \int_s^\infty \frac{t^{\kappa - 2}}{1 + t^\kappa} \, \mathrm{d}t. \quad (13)$$

(This makes $m_\kappa(0)$ well-defined, via the equation $s_\kappa^* = m_\kappa(0)^{1/\kappa}/\alpha$, and also verifies its positivity.)

The derivative of $q_\kappa(s, \alpha)$ with respect to $s$ is

$$\frac{\partial q_\kappa(s, \alpha)}{\partial s} = \frac{(\alpha - \beta)s^\kappa - \beta}{s^2(1 + s^\kappa)}. \quad (14)$$

Hence, since $\alpha > \beta$, we know the function $q_\kappa(s, \alpha)$ is strictly decreasing on $s \in (0, (\frac{\beta}{\alpha - \beta})^{1/\kappa}]$ and strictly increasing on $s \in [(\frac{\beta}{\alpha - \beta})^{1/\kappa}, \infty)$. Furthermore, $q_\kappa(s, \alpha) \to \infty$ as $s \to 0$ and $q_\kappa(s, \alpha) \to 0$ as $s \to \infty$. Hence, by the continuity of $s \mapsto q_\kappa(s, \alpha)$, we conclude that $q_\kappa(s, \alpha) = 0$ has a unique solution $s_\kappa^*$.

Using the chain rule, we can also show that $m_\kappa'(0)$ is well-defined, and that its value is given by

$$m_\kappa'(0) = \kappa \beta m_\kappa^2(0) \cdot (1 + (s_\kappa^*)^\kappa) / \left( \beta + (\beta - \alpha)(s_\kappa^*)^\kappa \right) > 0.$$

We leave the details to Appendix B.1.

Our next goal is to prove $\mathbb{E}_{\boldsymbol{\theta}}[\text{Error}] \overset{\text{P}}{\to} \mathcal{R}_\kappa(\alpha)$. Since $\alpha > \beta$, we have $p > n$ for large enough $N$. In this case,

$$\begin{aligned} \text{Error} &= \mathbb{E}_{\boldsymbol{x}, y}[(y - \boldsymbol{x}^\top \hat{\boldsymbol{\theta}})^2] = \mathbb{E}_{\boldsymbol{x}, y}[(\boldsymbol{x}_P^\top (\hat{\boldsymbol{\theta}}_P - \boldsymbol{\theta}_P) - \boldsymbol{x}_{P^c}^\top \boldsymbol{\theta}_{P^c})^2] \\ &= \|\boldsymbol{\Sigma}_P^{1/2}((\boldsymbol{\Pi}_{\boldsymbol{X}_P} - \boldsymbol{I})\boldsymbol{\theta}_P + \boldsymbol{X}_P^\top (\boldsymbol{X}_P \boldsymbol{X}_P^\top)^{-1} \boldsymbol{X}_{P^c} \boldsymbol{\theta}_{P^c})\|^2 + \|\boldsymbol{\Sigma}_{P^c}^{1/2} \boldsymbol{\theta}_{P^c}\|^2, \end{aligned}$$

where $\boldsymbol{\Pi}_{\boldsymbol{X}_P} := \boldsymbol{X}_P^\top (\boldsymbol{X}_P \boldsymbol{X}_P^\top)^{-1} \boldsymbol{X}_P$, and the diagonal matrices $\boldsymbol{\Sigma}_P$ and $\boldsymbol{\Sigma}_{P^c}$ are as defined in Section 2.2. Hence, $\mathbb{E}_{\boldsymbol{\theta}}[\text{Error}]$ is equal to

$$\underbrace{\text{tr}(\boldsymbol{\Sigma}_P(\boldsymbol{I} - \boldsymbol{\Pi}_{\boldsymbol{X}_P}))}_{\text{part 1}} + \underbrace{\text{tr}(\boldsymbol{X}_{P^c}^\top (\boldsymbol{X}_P \boldsymbol{X}_P^\top)^{-1} \boldsymbol{X}_P \boldsymbol{\Sigma}_P \boldsymbol{X}_P^\top (\boldsymbol{X}_P \boldsymbol{X}_P^\top)^{-1} \boldsymbol{X}_{P^c}) + \text{tr}(\boldsymbol{\Sigma}_{P^c})}_{\text{part 2}}. \quad (15)$$

We claim that

$$\text{part 1} \overset{\text{P}}{\to} \frac{N^{1 - \kappa} \beta}{m_\kappa(0)}, \quad \text{and} \quad \text{part 2} \overset{\text{P}}{\to} N^{1 - \kappa} \cdot \frac{m_\kappa'(0)}{m_\kappa^2(0)} \cdot \int_\alpha^1 t^{\kappa - 2} \, \mathrm{d}t + o_{\text{p}}(N^{1 - \kappa}); \quad (16)$$

together, they complete the proof that $\mathbb{E}_{\boldsymbol{\theta}}[\text{Error}] \overset{\text{P}}{\to} \mathcal{R}_\kappa(\alpha)$. Rigorous proofs of the claims in (16) are presented in Appendix B.2 and Appendix B.3; here, we give a heuristic argument that conveys the main idea. For part 1, let $\tilde{\boldsymbol{\Sigma}}_P = N^\kappa \boldsymbol{\Sigma}_P$ and $\tilde{\boldsymbol{X}}_P = N^{\kappa/2} \boldsymbol{X}_P$. This scaling ensures that the empirical eigenvalue distribution of $\tilde{\boldsymbol{\Sigma}}_P$ has a limiting distribution with probability density

$$f_\kappa(s) = \frac{1}{\kappa \alpha} s^{-1 - 1/\kappa} \cdot \mathbb{1}_{\{s \in [\alpha^{-\kappa}, \infty)\}}$$

(Lemma 2 in Appendix B.2). Also, under this scaling, we have

$$\begin{aligned} \text{tr}(\boldsymbol{\Sigma}_P(\boldsymbol{I} - \boldsymbol{\Pi}_{\boldsymbol{X}_P})) &= \lim_{\mu \to 0} \frac{n}{N^\kappa} \left( \frac{1}{n} \text{tr}(\tilde{\boldsymbol{\Sigma}}_P) - \frac{1}{n} \text{tr}(\tilde{\boldsymbol{\Sigma}}_P(\tilde{\boldsymbol{X}}_P^\top \tilde{\boldsymbol{X}}_P + \mu n \boldsymbol{I})^{-1} \tilde{\boldsymbol{X}}_P^\top \tilde{\boldsymbol{X}}_P) \right) \\ &= \lim_{\mu \to 0} \frac{n}{N^\kappa} \cdot \frac{\mu}{n} \text{tr}\left( \tilde{\boldsymbol{\Sigma}}_P \left( \frac{1}{n} \tilde{\boldsymbol{X}}_P^\top \tilde{\boldsymbol{X}}_P + \mu \boldsymbol{I} \right)^{-1} \right) = \lim_{\mu \to 0} \frac{n}{N^\kappa} \cdot \frac{\mu}{n} \text{tr}(\tilde{\boldsymbol{\Sigma}}_P \tilde{\boldsymbol{S}}_n), \quad (17) \end{aligned}$$

where $\tilde{\boldsymbol{S}}_n := (n^{-1}\tilde{\boldsymbol{X}}_P^\top \tilde{\boldsymbol{X}}_P + \mu\boldsymbol{I})^{-1}$. As long as the empirical eigenvalue distribution of $\tilde{\boldsymbol{\Sigma}}_P$ has a limiting distribution with bounded support, we have

$$\forall \mu > 0, \quad \mu \cdot \frac{1}{n}\operatorname{tr}\left(\tilde{\boldsymbol{\Sigma}}_P\left(\frac{1}{n}\tilde{\boldsymbol{X}}_P^\top \tilde{\boldsymbol{X}}_P + \mu\boldsymbol{I}\right)^{-1}\right) \xrightarrow{\mathrm{P}} \frac{1}{m_\kappa(-\mu)}, \tag{18}$$

where $m_\kappa(z)$ is, in fact, the Stieltjes transform of the limiting empirical eigenvalue distribution of $n^{-1}\tilde{\boldsymbol{X}}_P\tilde{\boldsymbol{X}}_P^\top$ (Lemma 1 in Appendix B.2); this follows from results of Dobriban and Wager [6], which in turn are derived from the results of Ledoit and Péché [11]. Assume we can exchange the two limits $\mu \to 0^+$ and $N \to \infty$, and also that (18) still holds for $f_\kappa(s)$ which has unbounded support. Then, from (17), we conclude

$$\text{part 1} \;=\; \operatorname{tr}(\boldsymbol{\Sigma}_P\left(\boldsymbol{I} - \boldsymbol{\Pi}_{\boldsymbol{X}_P}\right)) \;\xrightarrow{\mathrm{P}}\; \frac{N^{1-\kappa}\beta}{m_\kappa(0)}.$$

For part 2, note that $\boldsymbol{X}_{P^c}$ is independent of $\boldsymbol{X}_P$. Thus, conditional on $\boldsymbol{X}_P$, part 2 is a sum of $N - p$ independent random variables. Therefore, using Markov inequality, we can show that

$$\text{part 2} \quad \xrightarrow{\mathrm{P}} \quad \mathbb{E}_{\boldsymbol{X}_{P^c}}[\text{part 2}] \;=\; \operatorname{tr}\left(\boldsymbol{\Sigma}_{P^c}\right) \cdot \left(\operatorname{tr}\left(\boldsymbol{\Sigma}_P\boldsymbol{X}_P^\top\left(\boldsymbol{X}_P\boldsymbol{X}_P^\top\right)^{-2}\boldsymbol{X}_P\right) + 1\right)$$

$$=\quad \operatorname{tr}\left(\boldsymbol{\Sigma}_{P^c}\right) \cdot \left(\lim_{\mu\to0}\operatorname{tr}\left(\tilde{\boldsymbol{\Sigma}}_P\left(\tilde{\boldsymbol{X}}_P^\top\tilde{\boldsymbol{X}}_P + \mu n\boldsymbol{I}\right)^{-1}\tilde{\boldsymbol{X}}_P^\top\tilde{\boldsymbol{X}}_P\left(\tilde{\boldsymbol{X}}_P^\top\tilde{\boldsymbol{X}}_P + \mu n\boldsymbol{I}\right)^{-1}\right) + 1\right)$$

$$=\quad \operatorname{tr}\left(\boldsymbol{\Sigma}_{P^c}\right) \cdot \left(\lim_{\mu\to0}\frac{1}{n}\operatorname{tr}\left(\tilde{\boldsymbol{\Sigma}}_P\tilde{\boldsymbol{S}}_n\right) - \frac{\mu}{n}\operatorname{tr}\left(\tilde{\boldsymbol{\Sigma}}_P\tilde{\boldsymbol{S}}_n^2\right) + 1\right). \tag{19}$$

Again, if we ignore the fact that the support of $f_\kappa(s)$ is unbounded and assume the limits of $\mu \to 0$ and $N \to \infty$ can be exchanged, then by Lemma 7.4 of Dobriban and Wager [6], we have

$$\text{part 2} \;\xrightarrow{\mathrm{P}}\; \operatorname{tr}\left(\boldsymbol{\Sigma}_{P^c}\right) \cdot \left(\lim_{\mu\to0}\frac{1}{n}\operatorname{tr}\left(\tilde{\boldsymbol{\Sigma}}_P\tilde{\boldsymbol{S}}_n\right) - \frac{\mu}{n}\operatorname{tr}\left(\tilde{\boldsymbol{\Sigma}}_P\tilde{\boldsymbol{S}}_n^2\right) + 1\right) \xrightarrow{\mathrm{P}} \operatorname{tr}\left(\boldsymbol{\Sigma}_{P^c}\right) \cdot \frac{m_\kappa'(0)}{m_\kappa^2(0)}. \tag{20}$$

A straightforward analysis of $\operatorname{tr}(\boldsymbol{\Sigma}_{P^c})$ (as in (10)) completes the analysis of part 2 of (16).

*Remark* 1. Although Theorem 2 should intuitively hold given the results of Dobriban and Wager [6], a careful and more involved argument is needed to deal with the facts that $\|\tilde{\boldsymbol{\Sigma}}_P\|_2 \to \infty$ (since $\|\boldsymbol{\Sigma}_P^{-1}\|_2 \to \infty$) and $\mu \to 0$. For example, standard techniques only imply $\frac{\mu}{n}\operatorname{tr}(\tilde{\boldsymbol{\Sigma}}_P\tilde{\boldsymbol{S}}_n) = O_{\mathrm{p}}(N^\kappa)$. However, we need the stronger bound $\frac{\mu}{n}\operatorname{tr}(\tilde{\boldsymbol{\Sigma}}_P\tilde{\boldsymbol{S}}_n) = O_{\mathrm{p}}(1)$ (e.g., Appendix B.2.2).

## 2.4   Proof of Theorem 3

Comparing the expression for $\mathcal{R}_\kappa(\alpha)$ in (7) at $\alpha = 1$ to the expression for $\mathcal{R}_\kappa(\alpha^*)$ in (5), we see that it suffices to prove $m_\kappa(0)^{1/\kappa} > \alpha^*$. Recall that in Section 2.3, we have proved $s_\kappa^* := m_\kappa(0)^{1/\kappa}$ is the unique solution of the equation $q_\kappa(s, 1) = 0$. Furthermore, using the expression for the derivative of $q_\kappa(s, 1)$ with respect to $s$ in (14), we know that $q(s, 1) > 0 \Rightarrow s < s_\kappa^*$. Thus, we only need to show $q_\kappa(\alpha^*, 1) > 0 = h_\kappa(\alpha^*)$, where the equality is due to the definition of $\alpha^*$ in Theorem 1. Note that by the definitions of the functions $q_\kappa$ and $h_\kappa$ in (3) and (13), we have

$$h_\kappa(s) \;=\; \frac{\beta}{s} - \int_s^1 t^{\kappa-2}\,\mathrm{d}t - 1 \;=\; q_\kappa(s, 1) + \int_s^\infty \frac{t^{\kappa-2}}{(1+t^\kappa)}\,\mathrm{d}t - \int_s^1 t^{\kappa-2}\,\mathrm{d}t - 1.$$

Furthermore, $h_\kappa(s) - q_\kappa(s, 1)$ is increasing in $s$:

$$\frac{\mathrm{d}\left(h_\kappa(s) - q_\kappa(s, 1)\right)}{\mathrm{d}s} \;=\; -\frac{s^{\kappa-2}}{(1+s^\kappa)} + s^{\kappa-2} \;=\; \frac{s^{2\kappa-2}}{1+s^\kappa} \;>\; 0.$$

Hence, for all for all $s \in (0, 1]$, we have

$$h_\kappa(s) - q_\kappa(s, 1) \;\leq\; h_\kappa(1) - q_\kappa(1, 1) \;=\; \int_1^\infty \frac{t^{\kappa-2}}{(1+t^\kappa)}\,\mathrm{d}t - 1$$

$$=\; \int_1^\infty \frac{t^{\kappa-2}}{(1+t^\kappa)}\,\mathrm{d}t - \int_1^\infty \frac{1}{t^2}\,\mathrm{d}t \;=\; -\int_1^\infty \frac{1}{t^2(1+t^\kappa)}\,\mathrm{d}t \;<\; 0.$$

Since $\alpha^* < \beta < 1$, we have $0 = h_\kappa(\alpha^*) < q_\kappa(\alpha^*, 1)$, and thus we have $s_\kappa^* > \alpha^*$.

By inspection of the expression for $\mathcal{R}_\kappa(\alpha)$ in (4), it is also clear that $\mathcal{R}_\kappa(\alpha)/\mathcal{R}_\kappa(1) \to \infty$ as $\alpha \to \beta^-$.

# 3 Analysis under general eigenvalue decay

In this section, we extend the results from Section 2 (with noise) to hold under a more general assumption on the eigenvalues of $\boldsymbol{\Sigma}$. To simplify calculations, we use a slightly different feature selection procedure that includes all components $j$ such that $\lambda_j \geq \nu_N$, so $p = \sum_{j=1}^{N} \mathbb{1}_{\{\lambda_j \geq \nu_N\}}$.

Instead of Assumptions **A.1** and **A.2**, we assume the following:

**B.1** $\|\boldsymbol{\Sigma}\|_2 \leq C$ for some constant $C > 0$. Also, there exists a positive sequence $(c_N)_{N \geq 1}$ such that the empirical eigenvalue distribution of $c_N \boldsymbol{\Sigma}$ converges as $N \to \infty$ to $F = (1 - \delta)F_0 + \delta F_1$, where $\delta \in (0, 1]$, $F_0$ is a point mass of 0, and $F_1$ has a continuous probability density $f$ supported on either $[\eta_1, \eta_2]$ or $[\eta_1, \infty)$ for some constants $\eta_1, \eta_2 > 0$.

**B.2** There exist constants $\nu > 0$ and $\beta \in (0, \delta)$ s.t. $\nu_N c_N \to \nu$ and $n/N \to \beta$ as $n, N \to \infty$.

The $c_N$ in Assumption **B.1** generalizes the $N^\kappa$ scaling introduced in the proof of Theorem 2. In fact, Assumption **B.1** is more general than the eigenvalue assumptions made by Dobriban and Wager [6] and Hastie et al. [8]: the eigenvalues of $\boldsymbol{\Sigma}$ could decrease smoothly ($\delta = 1$), or there could be a sudden drop between (say) $\lambda_j$ and $\lambda_{j+1}$ ($\delta < 1$). Since $p$ is now determined by $\nu$, whether $p < n$ or $p > n$ is now determined by whether $\nu > \nu_b$ or $\nu < \nu_b$, where $\nu_b > \eta_1$ is given by the equation $\delta \int_{\nu_b}^{\infty} f(t) \, \mathrm{d}t = \beta$. Finally, by Assumption **B.1**,

$$\frac{p}{N} = \frac{1}{N} \sum_{j=1}^{N} \mathbb{1}_{\{c_N \lambda_j \geq \nu\}} \overset{\text{a.s.}}{\to} \delta \mathbb{E}_{s \sim f}[\mathbb{1}_{\{s \geq \nu\}}] = \delta \int_{\nu}^{\infty} f(t) \, \mathrm{d}t =: \alpha(\nu), \quad \forall \nu > 0. \quad (21)$$

For $\nu = 0$, i.e., $\nu_N = o(1/c_N)$, we choose $\nu_N$ be the $\delta N$ largest eigenvalues of $\boldsymbol{\Sigma}$, then $\alpha(\nu) = \delta$. Hence, combined with Assumption **B.2**, we have the same asymptotics considered in Section 2, except that $\beta$ is now restricted in $(0, \delta)$. This restriction on $\beta$ is required, otherwise both $c_N \boldsymbol{X}^\top \boldsymbol{X}$ and $c_N \boldsymbol{X} \boldsymbol{X}^\top$ are asymptotically singular.

The following theorem generalizes the results in Section 2 to hold under Assumptions **B.1** and **B.2**.

**Theorem 4.** *Assume **B.1** with sequence $(c_N)_{N \geq 1}$ and constants $C$, $\delta$, $\eta_1$, and $\eta_2$; and **B.2** with constants $\nu$ and $\beta$.*

(i) *Assume $\nu \in (\nu_b, \infty)$. Then*

$$\mathbb{E}_{\boldsymbol{w}, \boldsymbol{\theta}}[\text{Error}] \overset{\text{P}}{\to} \left( \frac{N}{c_N} \cdot \delta \int_{\eta_1}^{\nu} t f(t) \, \mathrm{d}t + \sigma^2 \right) \frac{\beta}{\beta - \delta \int_{\nu}^{\infty} f(t) \, \mathrm{d}t} =: \mathcal{R}_f(\nu, \sigma). \quad (22)$$

*Define $h_f(\nu) := \nu\beta - \nu\delta \int_{\nu}^{\infty} f(t) \, \mathrm{d}t - \delta \int_{\eta_1}^{\nu} t f(t) \, \mathrm{d}t$. If the equation $h_f(\nu) = 0$ has a solution on $(\nu_b, \infty) \bigcap \text{supp}(f)$, then the solution $\nu^*$ is unique, and*

$$\mathcal{R}_f(\nu^*, 0) = \min_{\nu \in (\nu_b, \infty)} \mathcal{R}_f(\nu, 0) = \frac{N\beta}{c_N} \cdot \nu^*. \quad (23)$$

*Otherwise,*

$$\inf_{\nu \in (\nu_b, \infty)} \mathcal{R}_f(\nu, 0) = \lim_{\nu \to \infty} \mathcal{R}_f(\nu, 0) = \frac{N}{c_N} \delta \int_{\eta_1}^{\infty} t f(t) \, \mathrm{d}t. \quad (24)$$

(ii) *Assume $\nu \in [0, \nu_b)$. Define $q_f(s, \nu) := s\beta - s\delta \int_{\nu}^{\infty} \frac{t f(t)}{s+t} \, \mathrm{d}t$. Then*

$$\mathbb{E}_{\boldsymbol{w}, \boldsymbol{\theta}}[\text{Error}] \overset{\text{P}}{\to} \frac{N\beta}{c_N} s_f^* + \beta \cdot \frac{\frac{N}{c_N} \delta \int_{\eta_1}^{\nu} t f(t) \, \mathrm{d}t + \sigma^2}{\delta s_f^* \int_{\nu}^{\infty} \frac{t f(t)}{(s_f^* + t)^2} \, \mathrm{d}t} =: \mathcal{R}_f(\nu, \sigma), \quad (25)$$

*where $s_f^*$ is the unique solution of the equation $q_f(s, \nu) = 0$.*

(iii) *Suppose $\sigma = 0$. Let $\nu^*$ be the minimizer of $\mathcal{R}_f(\nu, 0)$ over the interval $(\nu_b, \infty]$ (including $\infty$). Let $\mathcal{R}_f(\eta_1, 0)$ be the risk achieved at $\nu = \eta_1$. Then $\limsup_N \mathcal{R}_f(\eta_1, 0)/\mathcal{R}_f(\nu^*, 0) < 1$.*

The proof of this theorem is presented in Appendix D.

## 4 Discussion

Our results confirm the emergence of the "double descent" risk curve in a natural setting with Gaussian design. As in previous works [e.g., 3, 8, 13], the shape emerges when there is a spike at the interpolation threshold ($p = n$), which is typically caused by a near-zero minimum eigenvalue of the empirical covariance matrix.

More importantly, however, our results shed light on when the minimum risk is achieved before or after the interpolation threshold in terms of the noise level and eigenvalues of the (population) covariance matrix. For instance, when the eigenvalues decay very slowly or not at all ($\kappa < 1$), a smaller risk is achieved after the interpolation threshold ($p > n$) than any point before ($p < n$). On the other hand, when the eigenvalues decay more quickly ($\kappa > 1$), a smaller risk is achieved in the $p > n$ regime only in the noiseless setting. In general, the $p < n$ regime yields a smaller risk when the noise dominates the error due to model misspecification. Providing a full characterization is an important direction for future research.

Finally, we point out that the PCR estimator we study is a non-standard "oracle" estimator because it generally requires knowledge of $\Sigma$. Although it can be plausibly implemented in a semi-supervised setting (by estimating $\Sigma$ very accurately using unlabeled data), a full analysis that accounts for estimation errors in $\Sigma$, or of a more standard PCR estimator, remains open. However, we note that the PCR estimator with $p = N$ can be implemented, and in our analysis, the dominance of the $p > n$ regime is always established at $p = N$. We believe that this should be true for the standard PCR estimator as well.

### Acknowledgments

This research was supported by NSF CCF-1740833, a Sloan Research Fellowship, a Google Faculty Award, and a Cheung-Kong Graduate School of Business Fellowship.

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
