[Supplementary Material · nips_2019-appendix.pdf]

## A  Proof of Theorem 1

The full proof for Theorem 1 is presented in this section. Since $\alpha < \beta$, we have $p < n$ hold for large enough $N$. Then, the least square estimate $\hat{\boldsymbol{\theta}}_P$ is given by $\left(\boldsymbol{X}_P^\top \boldsymbol{X}_P\right)^{-1}\boldsymbol{X}_P^\top \boldsymbol{X}\boldsymbol{\theta}$ and the prediction error is given by

$$
\begin{aligned}
\text{Error} &= \mathbb{E}_{\boldsymbol{x},y}[(y - \boldsymbol{x}^\top \hat{\boldsymbol{\theta}})^2] = \mathbb{E}_{\boldsymbol{x},y}[(\boldsymbol{x}_P^\top(\boldsymbol{\theta}_P - \hat{\boldsymbol{\theta}}_P) + \boldsymbol{x}_{P^c}^\top \boldsymbol{\theta}_{P^c})^2] \\
&= \|\boldsymbol{\Sigma}_P^{1/2}(\boldsymbol{X}_P^\top \boldsymbol{X}_P)^{-1}\boldsymbol{X}_P^\top \boldsymbol{X}_{P^c}\boldsymbol{\theta}_{P^c}\|^2 + \|\boldsymbol{\Sigma}_{P^c}^{1/2}\boldsymbol{\theta}_{P^c}\|^2,
\end{aligned}
$$

where $\boldsymbol{\Sigma}_P \in \mathbb{R}^{p\times p}$ and $\boldsymbol{\Sigma}_{P^c} \in \mathbb{R}^{(N-p)\times(N-p)}$ are the two diagonal matrices whose diagonal elements are the first $p$ and last $N - p$ diagonal elements of $\boldsymbol{\Sigma}$ respectively. By our assumption on $\boldsymbol{\theta}$, we have

$$
\mathbb{E}_{\boldsymbol{\theta}}[\text{Error}] = \text{tr}(\boldsymbol{X}_{P^c}^\top \boldsymbol{X}_P(\boldsymbol{X}_P^\top \boldsymbol{X}_P)^{-1}\boldsymbol{\Sigma}_P(\boldsymbol{X}_P^\top \boldsymbol{X}_P)^{-1}\boldsymbol{X}_P^\top \boldsymbol{X}_{P^c}) + \text{tr}(\boldsymbol{\Sigma}_{P^c}).
$$

Our next step is to apply Markov inequality to show (4). Note that $\boldsymbol{X}_{P^c}$ is independent of $\boldsymbol{X}_P$. Hence, the expectation of Error given $\boldsymbol{X}_P$ is the following:

$$
\begin{aligned}
\mathbb{E}[\text{Error} \mid \boldsymbol{X}_P] &= \text{tr}(\boldsymbol{\Sigma}_{P^c}) \cdot (\text{tr}((\boldsymbol{X}_P^\top \boldsymbol{X}_P)^{-1}\boldsymbol{\Sigma}_P) + 1) \\
&= \text{tr}(\boldsymbol{\Sigma}_{P^c}) \cdot (\text{tr}((\bar{\boldsymbol{X}}_P^\top \bar{\boldsymbol{X}}_P)^{-1}) + 1),
\end{aligned} \tag{26}
$$

where $\bar{\boldsymbol{X}}_P = \boldsymbol{X}_P\boldsymbol{\Sigma}_P^{-\frac{1}{2}}$. (The expectation only conditions on $\boldsymbol{X}_P$; in particular, it averages over $\boldsymbol{X}_{P^c}$.) Further, the variance of Error given $\boldsymbol{X}_P$ is the following: letting $\boldsymbol{z} \sim \mathcal{N}(\boldsymbol{0},\boldsymbol{I})$,

$$
\begin{aligned}
&\text{var}(\text{Error} \mid \boldsymbol{X}_P) \\
&= \text{tr}(\boldsymbol{\Sigma}_{P^c}^2)\,\text{var}(\boldsymbol{z}^\top \boldsymbol{X}_P(\boldsymbol{X}_P^\top \boldsymbol{X}_P)^{-1}\boldsymbol{\Sigma}_P(\boldsymbol{X}_P^\top \boldsymbol{X}_P)^{-1}\boldsymbol{X}_P^\top \boldsymbol{z} \mid \boldsymbol{X}_P) \\
&\leq 2\,\text{tr}(\boldsymbol{\Sigma}_{P^c}^2)\|\boldsymbol{X}_P(\boldsymbol{X}_P^\top \boldsymbol{X}_P)^{-1}\boldsymbol{\Sigma}_P(\boldsymbol{X}_P^\top \boldsymbol{X}_P)^{-1}\boldsymbol{X}_P^\top\|_F^2 \\
&= 2\,\text{tr}(\boldsymbol{\Sigma}_{P^c}^2)\,\text{tr}((\boldsymbol{X}_P^\top \boldsymbol{X}_P)^{-1}\boldsymbol{\Sigma}_P(\boldsymbol{X}_P^\top \boldsymbol{X}_P)^{-1}\boldsymbol{\Sigma}_P) \\
&= 2\,\text{tr}(\boldsymbol{\Sigma}_{P^c}^2)\,\text{tr}((\bar{\boldsymbol{X}}_P^\top \bar{\boldsymbol{X}}_P)^{-2}).
\end{aligned}
$$

Hence, by Markov's inequality and the fact that $\text{tr}(\boldsymbol{\Sigma}_{P^c}^2) \leq \text{tr}(\boldsymbol{\Sigma}_{P^c})^2$, we have

$$
\mathbb{E}_{\boldsymbol{\theta}}[\text{Error}] = \mathbb{E}[\text{Error}|\boldsymbol{X}_P] \cdot \left(1 + O_{\text{p}}\left(\text{tr}((\bar{\boldsymbol{X}}_P^\top \bar{\boldsymbol{X}}_P)^{-2})^{1/2} \cdot \left(\text{tr}((\bar{\boldsymbol{X}}_P^\top \bar{\boldsymbol{X}}_P)^{-1}) + 1\right)^{-1}\right)\right). \tag{27}
$$

Our next step is to simplify (27). Note that $\bar{\boldsymbol{X}}_P$ is a standard Gaussian matrix. Hence, when $\alpha > 0$, from (2.104) and (2.105) of [18], we know

$$
\frac{n}{p}\text{tr}\left(\left(\bar{\boldsymbol{X}}_P^\top \bar{\boldsymbol{X}}_P\right)^{-1}\right) \overset{\text{a.s.}}{\to} \frac{\beta}{\beta - \alpha} \quad \text{and} \quad \frac{n^2}{p}\cdot\text{tr}\left(\left(\bar{\boldsymbol{X}}_P^\top \bar{\boldsymbol{X}}_P\right)^{-2}\right) \overset{\text{a.s.}}{\to} \frac{\beta^3}{(\beta - \alpha)^3}.
$$

When $\alpha = 0$, i.e., $p = o(n)$, from (2.110) and (2.111) of [18], we know

$$
\frac{n}{p}\text{tr}\left(\left(\bar{\boldsymbol{X}}_P^\top \bar{\boldsymbol{X}}_P\right)^{-1}\right) \overset{\text{a.s.}}{\to} 1 \quad \text{and} \quad \frac{n^2}{p}\text{tr}\left(\left(\bar{\boldsymbol{X}}_P^\top \bar{\boldsymbol{X}}_P\right)^{-2}\right) \overset{\text{a.s.}}{\to} 1.
$$

Therefore, with (26) and (27), we have for all $\alpha < \beta$,

$$
\begin{aligned}
\mathbb{E}_{\boldsymbol{\theta}}[\text{Error}] &= \mathbb{E}[\text{Error}|\boldsymbol{X}_P] \cdot \left(1 + O_{\text{p}}\left(\sqrt{\frac{\beta\alpha(\beta - \alpha)^{-3}}{N\left(\frac{\alpha}{\beta-\alpha} + 1\right)^2}}\right)\right) \\
&= \mathbb{E}[\text{Error}|\boldsymbol{X}_P] \cdot \left(1 + O_{\text{p}}\left(\frac{1}{\sqrt{N}}\right)\right) \\
&\overset{\text{P}}{\to} \text{tr}(\boldsymbol{\Sigma}_{P^c}) \cdot \left(\text{tr}\left(\left(\bar{\boldsymbol{X}}_P^\top \bar{\boldsymbol{X}}_P\right)^{-1}\right) + 1\right) \overset{\text{P}}{\to} \text{tr}(\boldsymbol{\Sigma}_{P^c}) \cdot \frac{\beta}{\beta - \alpha}. \tag{28}
\end{aligned}
$$

Our final step is to analyze $\text{tr}(\boldsymbol{\Sigma}_{P^c})$. Note that $\int_s^{s+1} t^{-\kappa}\,\text{d}t < \frac{1}{s^\kappa} < \int_{s-1}^s t^{-\kappa}\,\text{d}t$. Hence, we have

$$
\int_{p+1}^N \frac{N^\kappa}{t^\kappa}\,\text{d}t/N < \frac{N^\kappa}{N}\sum_{i=p+1}^N \frac{1}{i^\kappa} = N^{\kappa-1}\text{tr}(\boldsymbol{\Sigma}_{P^c}) < \int_p^N \frac{N^\kappa}{t^\kappa}\,\text{d}t/N. \tag{29}
$$

Therefore, we know $\text{tr}(\boldsymbol{\Sigma}_{P^c}) \to N^{1-\kappa}\int_\alpha^1 t^{-\kappa}\,\text{d}t$ as $p \to \infty$ and thus (4) holds.

## B Proof of Theorem 2

### B.1 Existence and positivity of $m'_\kappa(0)$

We already showed in Section 2.3 that $m_\kappa(0)$ is well-defined. We now show that $m_\kappa(z)$ is well-defined in a neighborhood of $z = 0$, which we can then use to establish the existence and positivity of $m'_\kappa(0)$. Note that, in fact, Lemma 1 in Appendix B.2 shows that $m_\kappa(z)$ is the Stieltjes transform of a distribution, specifically the limiting distribution of the empirical eigenvalue distribution of $\Sigma_P$. This lemma, which is proved in Appendix B.4, establishes the existence of the Stieltjes transform for all $z \le 0$. Here, we just give the arguments needed to show the existence of $m'_\kappa(0)$.

Define

$$z_\kappa(m) := -\frac{1}{m} + \frac{1}{\beta} \int_{\alpha^{-\kappa}}^{\infty} \frac{1}{\kappa t^{1/\kappa}(1 + t \cdot m)}\, \mathrm{d}t.$$

Based on (6), we can consider $z_\kappa(m)$ to be the inverse of $m_\kappa(z)$ wherever $m_\kappa(z)$ exists. Then, note that

$$\frac{\mathrm{d}z_\kappa(m)}{\mathrm{d}m} = \frac{1}{m^2} - \frac{1}{\beta} \int_{\alpha^{-\kappa}}^{\infty} \frac{t^2}{\kappa t^{1+1/\kappa}(1 + t \cdot m)^2}\, \mathrm{d}t.$$

Hence, we have

$$\frac{\mathrm{d}z_\kappa(m)}{\mathrm{d}m} \gtrless 0 \quad \Leftrightarrow \quad 1 \gtrless \frac{1}{\beta} \int_{\alpha^{-\kappa}}^{\infty} \frac{t^2}{\kappa t^{1+1/\kappa}(m^{-1} + t)^2}\, \mathrm{d}t.$$

Note that $\frac{1}{\beta} \int_{\alpha^{-\kappa}}^{\infty} \frac{t^2}{\kappa t^{1+1/\kappa}(m^{-1}+t)^2}\, \mathrm{d}t$ is a increasing function of $m$ with

$$\frac{1}{\beta} \int_{\alpha^{-\kappa}}^{\infty} \frac{t^2}{\kappa t^{1+1/\kappa}(m^{-1} + t)^2}\, \mathrm{d}t \quad \to \quad 0 \quad \text{as } m \to 0;$$

$$\frac{1}{\beta} \int_{\alpha^{-\kappa}}^{\infty} \frac{t^2}{\kappa t^{1+1/\kappa}(m^{-1} + t)^2}\, \mathrm{d}t \quad \to \quad \frac{1}{\beta} > 1 \quad \text{as } m \to \infty.$$

Hence, there exists a constant $m_c$ such that for all $0 < m < m_c$, the function $z_\kappa(m)$ is increasing on the interval $(0, m_c)$ and decreasing on $(m_c, \infty)$. Furthermore, note that

$$m \cdot z_\kappa(m) = \frac{1}{\beta} \int_{\alpha^{-\kappa}}^{\infty} \frac{1}{\kappa t^{1/\kappa}(m^{-1} + t)}\, \mathrm{d}t - 1. \tag{30}$$

Evaluating this integral as $m \to 0^+$ and as $m \to +\infty$ shows that

$$m \cdot z_\kappa(m) \to \begin{cases} -1 & \text{as } m \to 0^+, \\ \frac{1}{\beta} - 1 > 0 & \text{as } m \to +\infty, \end{cases} \tag{31}$$

which in turn implies

$$z_\kappa(m) \to \begin{cases} -\infty & \text{as } m \to 0^+, \\ 0 & \text{as } m \to +\infty. \end{cases} \tag{32}$$

Therefore, $z_\kappa(m)$ is strictly increasing on $z \le 0$. Further, for $z \in [0, z_\kappa(m_c)]$, there are two only solutions of $m$ satisfying (6). Therefore, since $m_\kappa(z)$ is defined to be the smallest positive solution of (6), the mapping between $z \in (-\infty, z_\kappa(m_c)]$ and $m \in (0, m_c]$ defined by $z_\kappa(m)$ and $m_\kappa(z)$ is continuous, one-to-one, and $z_\kappa(m_c) > 0$. This shows that $m_\kappa(z)$ is well-defined and continuous at $z = 0$. Then, by continuity of the defining expression, we conclude that $m'_\kappa(0)$ exists.

Next, we use the chain rule to calculate the value of $m'_\kappa(0)$. From the definition of $m_\kappa$ in (6), the change-of-variable in (12), and the definition of $q_\kappa$ in (13), we have

$$-z = \frac{1}{\beta\alpha \cdot m_\kappa(z)^{1-1/\kappa}} \cdot q_\kappa\left(\frac{m_\kappa(z)^{1/\kappa}}{\alpha}, \alpha\right) \tag{33}$$

for $z$ in a neighborhood of $z = 0$. Also, from the analysis in Section 2.3, we have $m_\kappa(0) = (s_\kappa^* \alpha)^\kappa$ and $q_\kappa(s_\kappa^*, \alpha) = 0$. Then, taking the derivative with respect to $z$ on both sides of (33) and with the chain rule, we have

$$
-1 = \left(\frac{1}{\kappa} - 1\right) \cdot \frac{m_\kappa(z)^{1/\kappa - 2}}{\beta\alpha} \cdot q_\kappa\left(\frac{m_\kappa(z)^{1/\kappa}}{\alpha}, \alpha\right)
$$
$$
+ \frac{1}{\beta\alpha m_\kappa(z)^{1-1/\kappa}} \cdot \frac{\partial q_\kappa(s, \alpha)}{\partial s}\bigg|_{s = \frac{m_\kappa(z)^{1/\kappa}}{\alpha}} \cdot \frac{m_\kappa(z)^{1/\kappa - 1}}{\kappa\alpha} \cdot m_\kappa'(z).
$$

Hence, plugging in $z = 0$ and solving for $m_\kappa'(0)$ gives

$$
m_\kappa'(0) = \frac{\kappa\beta\alpha^2(m_\kappa(0))^{2-2/\kappa}}{-\frac{\partial q_\kappa(s,\alpha)}{\partial s}\big|_{s = s_\kappa^*}}
$$

Then, using the formula for the derivative of $q_\kappa$ in (14), we have

$$
m_\kappa'(0) = \kappa\beta m_\kappa^2(0) \cdot \frac{1 + (s_\kappa^*)^\kappa}{\beta - (\alpha - \beta)(s_\kappa^*)^\kappa}. \tag{34}
$$

Since $(s_\kappa^*)^\kappa < \beta/(\alpha - \beta)$ (recall the argument in Section 2.3 following Equation (14)), it follows that $m_\kappa'(0) > 0$.

## B.2 Analysis of part 1

In this section, we will prove that

$$
\mathrm{tr}\left(\mathbf{\Sigma}_P\left(\mathbf{I} - \mathbf{\Pi}_{\mathbf{X}_P}\right)\right) \xrightarrow{\mathrm{P}} \frac{N^{1-\kappa}\beta}{m_\kappa(0)}. \tag{35}
$$

(The existence and uniqueness of $m_\kappa^* := m_\kappa(0)$ is proved in the beginning of Section 2.3.) Let $\tilde{\mathbf{\Sigma}}_P = N^\kappa \mathbf{\Sigma}_P$ and $\tilde{\mathbf{X}}_P = N^{\kappa/2} \mathbf{X}_P$, then we have, for all $\mu > 0$,

$$
\mathrm{tr}\left(\mathbf{\Sigma}_P\left(\mathbf{I} - \mathbf{\Pi}_{\mathbf{X}_P}\right)\right) = \frac{n}{N^\kappa}\left(\frac{1}{n}\mathrm{tr}\left(\tilde{\mathbf{\Sigma}}_P\right) - \frac{1}{n}\mathrm{tr}\left(\tilde{\mathbf{\Sigma}}_P\tilde{\mathbf{X}}_P^\top\left(\tilde{\mathbf{X}}_P\tilde{\mathbf{X}}_P^\top\right)^{-1}\tilde{\mathbf{X}}_P\right)\right)
$$
$$
= \frac{n}{N^\kappa}\left(\frac{1}{n}\mathrm{tr}\left(\tilde{\mathbf{\Sigma}}_P\right) - \frac{1}{n}\mathrm{tr}\left(\tilde{\mathbf{\Sigma}}_P\left(\tilde{\mathbf{X}}_P^\top\tilde{\mathbf{X}}_P + \mu n\mathbf{I}\right)^{-1}\tilde{\mathbf{X}}_P^\top\tilde{\mathbf{X}}_P\right) + \epsilon_{\mu_n}\right)
$$
$$
= \frac{n}{N^\kappa}\left(\mu \cdot \frac{1}{n}\mathrm{tr}\left(\tilde{\mathbf{\Sigma}}_P\left(\frac{1}{n}\tilde{\mathbf{X}}_P^\top\tilde{\mathbf{X}}_P + \mu\mathbf{I}\right)^{-1}\right) + \epsilon_{\mu_n}\right), \tag{36}
$$

where $\epsilon_{\mu_n}$ is given by

$$
\epsilon_{\mu_n} := \frac{1}{n}\mathrm{tr}\left(\tilde{\mathbf{\Sigma}}_P\left(\tilde{\mathbf{X}}_P^\top\tilde{\mathbf{X}}_P + n\mu\mathbf{I}\right)^{-1}\tilde{\mathbf{X}}_P^\top\tilde{\mathbf{X}}_P\right) - \frac{1}{n}\mathrm{tr}\left(\tilde{\mathbf{\Sigma}}_P\tilde{\mathbf{X}}_P^\top\left(\tilde{\mathbf{X}}_P\tilde{\mathbf{X}}_P^\top\right)^{-1}\tilde{\mathbf{X}}_P\right).
$$

Since $n/N^\kappa \to N^{1-\kappa}\beta$, the claim in (35) is implied by

$$
\mu \cdot \frac{1}{n}\mathrm{tr}\left(\tilde{\mathbf{\Sigma}}_P\left(\frac{1}{n}\tilde{\mathbf{X}}_P^\top\tilde{\mathbf{X}}_P + \mu\mathbf{I}\right)^{-1}\right) + \epsilon_{\mu_n} = \frac{1}{m_\kappa(0)} + o_\mathrm{p}(1).
$$

Hence, our task is reduced to finding a suitable positive sequence $(\mu_n)_{n \geq 1}$ such that the following hold:

$$
|\epsilon_{\mu_n}| = o_\mathrm{p}(1), \tag{37}
$$

and

$$
\mu_n \cdot \frac{1}{n}\mathrm{tr}\left(\tilde{\mathbf{\Sigma}}_P\left(\frac{1}{n}\tilde{\mathbf{X}}_P^\top\tilde{\mathbf{X}}_P + \mu_n\mathbf{I}\right)^{-1}\right) \xrightarrow{\mathrm{P}} \frac{1}{m_\kappa(0)}. \tag{38}
$$

With foresight, we shall assume that

$$
\mu_n < \min\left\{\frac{1}{\sqrt{N}}, o(N^{-\kappa})\right\}.
$$

 **B.2.1  Proof of Equation** (37)

347  Let us first show (37). Towards this end, we bound $|\epsilon_{\mu_n}|$ as follows:

$$
\begin{aligned}
|\epsilon_{\mu_n}| &= \frac{1}{n}\left|\operatorname{tr}\left(\tilde{\boldsymbol{\Sigma}}_P\left(\left(\tilde{\boldsymbol{X}}_P^\top\tilde{\boldsymbol{X}}_P + \mu_n n\boldsymbol{I}\right)^{-1}\tilde{\boldsymbol{X}}_P^\top\tilde{\boldsymbol{X}}_P - \tilde{\boldsymbol{X}}_P^\top\left(\tilde{\boldsymbol{X}}_P\tilde{\boldsymbol{X}}_P^\top\right)^{-1}\tilde{\boldsymbol{X}}_P\right)\right)\right| \\[2mm]
&\overset{(i)}{\leq} \frac{1}{n}\|\tilde{\boldsymbol{\Sigma}}_P\|_2\operatorname{tr}\left(\tilde{\boldsymbol{X}}_P^\top\left(\tilde{\boldsymbol{X}}_P\tilde{\boldsymbol{X}}_P^\top\right)^{-1}\tilde{\boldsymbol{X}}_P - \left(\tilde{\boldsymbol{X}}_P^\top\tilde{\boldsymbol{X}}_P + \mu_n n\boldsymbol{I}\right)^{-1}\tilde{\boldsymbol{X}}_P^\top\tilde{\boldsymbol{X}}_P\right) \\[2mm]
&\leq \frac{N^\kappa}{n}\cdot\sum_{i=1}^n\frac{\mu_n}{\tilde{\lambda}_i + \mu_n} = N^\kappa\cdot\mu_n\cdot m_n(-\mu_n) \leq N^\kappa\cdot\frac{\mu_n}{\min_i(\tilde{\lambda}_i)},
\end{aligned}
\tag{39}
$$

348  where $\tilde{\lambda}_i$ is the $i$-th eigenvalue of $\frac{1}{n}\tilde{\boldsymbol{X}}_P\tilde{\boldsymbol{X}}_P^\top$ and $m_n(z)$ is the Stieltjes transform of the empirical
349  eigenvalue distribution of $\frac{1}{n}\tilde{\boldsymbol{X}}\tilde{\boldsymbol{X}}^\top$. Inequality (i) holds because

$$
\tilde{\boldsymbol{X}}_P^\top\left(\tilde{\boldsymbol{X}}_P\tilde{\boldsymbol{X}}_P^\top\right)^{-1}\tilde{\boldsymbol{X}}_P - \left(\tilde{\boldsymbol{X}}_P^\top\tilde{\boldsymbol{X}}_P + \mu_n n\boldsymbol{I}\right)^{-1}\tilde{\boldsymbol{X}}_P^\top\tilde{\boldsymbol{X}}_P
$$

350  is positive semi-definite. Hence, the proof of (37) only require us to lower bound $\min_i(\tilde{\lambda}_i)$ and the
351  following lemma will help us complete this task.

352  **Lemma 1.** *Suppose the empirical eigenvalue distribution of the diagonal matrix $\boldsymbol{H}$ converges to*
353  *a limiting distribution $\mathcal{H}$ with probability density function $f_h$. Assume that the support of $f_h$ is a*
354  *subset of the interval $[\eta_1, \infty)$ for some positive constant $\eta_1$. Let $\bar{\boldsymbol{X}} \in \mathbb{R}^{n\times p}$ be a standard Gaussian*
355  *matrix and suppose $p/n \to \gamma > 1$. Let $m_n(z)$ be the Stieltjes transform of the empirical eigenvalue*
356  *distribution $\mathcal{F}_n$ of $\frac{1}{n}\bar{\boldsymbol{X}}\boldsymbol{H}\bar{\boldsymbol{X}}^\top$. Then $\mathcal{F}_n$ converges to a limit $\mathcal{F}$ whose Stieltjes transform, denoted by*
357  *$m(z)$, satisfies*

$$
m(z) = -\left(z - \gamma\int_{\eta_1}^\infty\frac{tf_h(t)\,\mathrm{d}t}{1 + t\cdot m(z)}\right)^{-1}, \quad \forall z \in \operatorname{supp}(\mathcal{F})^c.
\tag{40}
$$

358  *Further, there exists a constant $c_\epsilon > 0$ such that the minimum eigenvalue of $\frac{1}{n}\bar{\boldsymbol{X}}\boldsymbol{H}\bar{\boldsymbol{X}}^\top$ is lower-*
359  *bounded by $c_\epsilon$ in probability. Finally, for any increasing sequence $z_n \to 0^-$, we have*

$$
m_n(z_n) \overset{\mathrm{P}}{\to} m(0) \quad and \quad m_n'(z_n) \overset{\mathrm{P}}{\to} m'(0).
\tag{41}
$$

360  The proof of Lemma 1 is shown in Appendix B.4. Hence to apply Lemma 1, we need the empirical
361  distribution of the eigenvalues of the covariance matrix $\boldsymbol{\Sigma}_P$ converges to a limiting distribution and
362  thus we need to scale $\boldsymbol{\Sigma}_P$ properly. The following lemma confirms that the correct scaling is $p^\kappa$.

363  **Lemma 2.** *Let $S = \{i\}_{p_1 < i \leq p_2}$ with $0 \leq p_1 < p_2 \leq N$. Suppose $\frac{p_1}{N} \to \alpha_1$ and $\frac{p_2}{N} \to \alpha_2$*
364  *with $0 \leq \alpha_1 < \alpha_2 \leq 1$. Then, the empirical eigenvalue distribution of $N^\kappa\boldsymbol{\Sigma}_S$ converges to a*
365  *(non-random) distribution $\mathcal{F}$ with probability density function $f$ given by*

$$
f(s) = \begin{cases} \dfrac{1}{\kappa(\alpha_2 - \alpha_1)}s^{-1-\frac{1}{\kappa}}\cdot\mathbb{1}_{\{s\in[\alpha_2^{-\kappa}, \alpha_1^{-\kappa}]\}}, & \alpha_1 > 0 \\[3mm] \dfrac{1}{\kappa\alpha_2}s^{-1-\frac{1}{\kappa}}\cdot\mathbb{1}_{\{s\in[\alpha_2^{-\kappa},\infty)\}}, & \alpha_1 = 0 \end{cases}.
\tag{42}
$$

366  The proof of Lemma 2 is shown in Appendix B.5. Using Lemma 1, Lemma 2, and (39), we see that
367  since $\mu_n = o(N^{-\kappa})$, we have

$$
|\epsilon_{\mu_n}| = o_{\mathrm{p}}(1),
$$

368  which establishes Equation (37).

369  **B.2.2  Proof of Equation** (38)

370  Our next goal is to prove (38), i.e.,

$$
\frac{\mu_n}{n}\operatorname{tr}(\tilde{\boldsymbol{\Sigma}}_P\tilde{\boldsymbol{S}}_n) \overset{\mathrm{P}}{\to} \frac{1}{m_\kappa(0)}
$$

 where

$$\tilde{S}_n := \left( \frac{1}{n} \tilde{X}_P^\top \tilde{X}_P + \mu_n I \right)^{-1}.$$

372 The same result has been proved in Lemma 2.2 of [11] with additional assumption that the empirical
373 eigenvalue distribution of $\tilde{\Sigma}$ converges to a limiting distribution with bounded support. However,
374 this assumption does not hold in our case. We employ a similar proof strategy with more involved
375 arguments based on leave-one-out estimates [19].

376 Let $\tilde{x}_i$ be the $i$-th row of $\tilde{X}_P$. Then using the identity

$$\tilde{S}_n^{-1} - \mu_n I = \frac{1}{n} \sum_{i=1}^{n} \tilde{x}_i \tilde{x}_i^\top,$$

377 we have

$$
\begin{aligned}
\frac{1}{n} \sum_{i=1}^{n} \tilde{x}_i^\top \tilde{S}_n \tilde{x}_i &= \frac{1}{n} \operatorname{tr} \left( \sum_{i=1}^{n} \tilde{S}_n \tilde{x}_i \tilde{x}_i^\top \right) \\
&= \operatorname{tr} \left( \tilde{S}_n (\tilde{S}_n^{-1} - \mu_n I) \right) \\
&= \operatorname{tr} \left( I - \mu_n \tilde{S}_n \right).
\end{aligned}
\tag{43}
$$

378 For each $i = 1, \ldots, n$, define

$$\tilde{S}_n^{\setminus i} := \left( \frac{1}{n} \tilde{X}_P^\top \tilde{X}_P - \frac{1}{n} \tilde{x}_i \tilde{x}_i^\top + \mu_n I \right)^{-1} = \left( \tilde{S}_n^{-1} - n^{-1} \tilde{x}_i \tilde{x}_i^\top \right)^{-1}.$$

379 By the Sherman-Morrison formula, we have

$$\tilde{S}_n = \tilde{S}_n^{\setminus i} - \frac{1}{n} \cdot \frac{\tilde{S}_n^{\setminus i} \tilde{x}_i \tilde{x}_i^\top \tilde{S}_n^{\setminus i}}{1 + \frac{1}{n} \tilde{x}_i^\top \tilde{S}_n^{\setminus i} \tilde{x}_i}. \tag{44}$$

380 Hence, with (43), we have

$$
\begin{aligned}
\operatorname{tr}(I - \mu_n \tilde{S}_n) &= \frac{1}{n} \sum_{i=1}^{n} \tilde{x}_i^\top \tilde{S}_n \tilde{x}_i = \frac{1}{n} \sum_{i=1}^{n} \tilde{x}_i^\top \left( \tilde{S}_n^{\setminus i} - \frac{1}{n} \cdot \frac{\tilde{S}_n^{\setminus i} \tilde{x}_i \tilde{x}_i^\top \tilde{S}_n^{\setminus i}}{1 + \frac{1}{n} \tilde{x}_i^\top \tilde{S}_n^{\setminus i} x_i} \right) \tilde{x}_i \\
&= \frac{1}{n} \sum_{i=1}^{n} \frac{\tilde{x}_i^\top \tilde{S}_n^{\setminus i} \tilde{x}_i}{1 + \frac{1}{n} \tilde{x}_i^\top \tilde{S}_n^{\setminus i} \tilde{x}_i} = n - \sum_{i=1}^{n} \frac{1}{1 + \frac{1}{n} \tilde{x}_i^\top \tilde{S}_n^{\setminus i} \tilde{x}_i}.
\end{aligned}
$$

381 Since $\operatorname{tr}(I - \mu_n \tilde{S}_n) = n - n \mu_n \cdot m_n(-\mu_n)$, we have

$$m_n(-\mu_n) = \frac{1}{n} \sum_{i=1}^{n} \frac{1}{\mu_n + \frac{\mu_n}{n} \tilde{x}_i^\top \tilde{S}_n^{\setminus i} \tilde{x}_i}. \tag{45}$$

382 Note that $|m_n(-\mu_n) - m_n(0)| \le \frac{\mu_n}{\min(\tilde{\lambda}_i^2)}$ where $\tilde{\lambda}_i$ is the $i$th eigenvalue of $\frac{1}{n} \tilde{X}_P \tilde{X}_P^\top$. By Lemma 1,
383 we have

$$m_n(-\mu_n) = m_n(0) + O_p(\mu_n) \overset{P}{\to} m_\kappa(0). \tag{46}$$

384 Therefore, the LHS of (45) converges to $m_\kappa(0)$ in probability. Then we just need to show the RHS of
385 (45) converges to

$$\left( \frac{\mu_n}{n} \operatorname{tr} \left( \tilde{\Sigma}_P \tilde{S}_n \right) \right)^{-1}$$

386 in probability. Let

$$\Delta_i := \frac{\mu_n}{n} \operatorname{tr} \left( \tilde{\Sigma}_P \tilde{S}_n \right) - \frac{\mu_n}{n} \tilde{x}_i^\top \tilde{S}_n^{\setminus i} \tilde{x}_i - \mu_n,$$

then note that

$$\left| \left( \frac{\mu_n}{n} \operatorname{tr}\left(\tilde{\boldsymbol{\Sigma}}_P \tilde{\boldsymbol{S}}_n\right)\right)^{-1} - m_n(-\mu_n)\right| = \left| \left( \frac{\mu_n}{n} \operatorname{tr}\left(\tilde{\boldsymbol{\Sigma}}_P \tilde{\boldsymbol{S}}_n\right)\right)^{-1} - \frac{1}{n}\sum_{i=1}^{n} \frac{1}{\mu_n + \frac{\mu_n}{n}\tilde{\boldsymbol{x}}_i^\top \tilde{\boldsymbol{S}}_n^{\backslash i}\tilde{\boldsymbol{x}}_i}\right|$$

$$= \left| \frac{1}{n}\sum_{i=1}^{n} \frac{\Delta_i}{\frac{\mu_n}{n}\operatorname{tr}\left(\tilde{\boldsymbol{\Sigma}}_P \tilde{\boldsymbol{S}}_n\right) \cdot \left( \frac{\mu_n}{n}\operatorname{tr}\left(\tilde{\boldsymbol{\Sigma}}_P \tilde{\boldsymbol{S}}_n\right) - \Delta_i\right)}\right|$$

$$\leq \sup_i \frac{|\Delta_i|}{\frac{\mu_n}{n}\operatorname{tr}\left(\tilde{\boldsymbol{\Sigma}}_P \tilde{\boldsymbol{S}}_n\right) \cdot \left| \frac{\mu_n}{n}\operatorname{tr}\left(\tilde{\boldsymbol{\Sigma}}_P \tilde{\boldsymbol{S}}_n\right) - |\Delta_i|\right|}.$$

We claim that

$$\frac{\mu_n}{n}\operatorname{tr}\left(\tilde{\boldsymbol{\Sigma}}_P \tilde{\boldsymbol{S}}_n\right) = \Theta_{\mathrm{p}}(1);$$

$$\sup_i |\Delta_i| = O_{\mathrm{p}}\left(\frac{\ln N}{\sqrt{N}}\right)$$

(Proposition 1 and Proposition 2 below). Then with (46), we have

$$\left( \frac{\mu_n}{n}\operatorname{tr}\left(\tilde{\boldsymbol{\Sigma}}_P \tilde{\boldsymbol{S}}_n\right)\right)^{-1} \xrightarrow{\mathrm{P}} m_\kappa(0).$$

This in turn implies Equation (38) as desired.

### B.2.3 Supporting propositions

**Proposition 1.**

$$\frac{\mu_n}{n}\operatorname{tr}\left(\tilde{\boldsymbol{\Sigma}}_P \tilde{\boldsymbol{S}}_n\right) = \Theta_{\mathrm{p}}(1).$$

*Proof.* Note that

$$\frac{\mu_n}{n}\operatorname{tr}\left(\tilde{\boldsymbol{\Sigma}}_P \tilde{\boldsymbol{S}}_n\right) \overset{(i)}{\geq} \frac{\mu_n}{n}\operatorname{tr}\left(\tilde{\boldsymbol{S}}_n\right) = \frac{\mu_n}{n}\operatorname{tr}\left( \left( \frac{1}{n}\tilde{\boldsymbol{X}}_P^\top \tilde{\boldsymbol{X}}_P + \mu_n \boldsymbol{I}\right)^{-1}\right)$$

$$\overset{(ii)}{\geq} \frac{\mu_n}{n}\cdot \frac{p-n}{\mu_n} \to \frac{\alpha-\beta}{\beta} > 0,$$

where inequality (i) holds due to the fact that $\tilde{\boldsymbol{\Sigma}}_P$ is a diagonal matrix with diagonal elements lower bounded by 1, and inequality (ii) holds due to the fact that $\left( \frac{1}{n}\tilde{\boldsymbol{X}}_P^\top \tilde{\boldsymbol{X}}_P + \mu_n \boldsymbol{I}\right)^{-1}$ has at least $p-n$ number of eigenvalues $\frac{1}{\mu_n}$. Hence, we have $\frac{\mu_n}{n}\operatorname{tr}(\tilde{\boldsymbol{\Sigma}}_P \tilde{\boldsymbol{S}}_n) = \Omega_{\mathrm{p}}(1)$. To show $\frac{\mu_n}{n}\operatorname{tr}\left(\tilde{\boldsymbol{\Sigma}}_P \tilde{\boldsymbol{S}}_n\right) = O_{\mathrm{p}}(1)$ as well, let us introduce $\bar{\boldsymbol{S}}_n = \tilde{\boldsymbol{\Sigma}}_P^{1/2}\boldsymbol{S}_n \tilde{\boldsymbol{\Sigma}}_P^{1/2}$, then we have

$$\frac{\mu_n}{n}\operatorname{tr}\left(\tilde{\boldsymbol{\Sigma}}_P \tilde{\boldsymbol{S}}_n\right) = \frac{\mu_n}{n}\operatorname{tr}\left(\bar{\boldsymbol{S}}_n\right) \leq \mu_n \frac{p}{n}\|\bar{\boldsymbol{S}}_n\|_2.$$

Therefore, as $p/n \to \alpha/\beta$, we just need to upper bound $\|\bar{\boldsymbol{S}}_n\|_2$. To do this, we use the following lemma.

**Lemma 3.** *Let $\boldsymbol{\Sigma} \in \mathbb{R}^{p\times p}$ be a diagonal matrix. Let $\bar{\boldsymbol{X}} \in \mathbb{R}^{n\times p}$ be a standard Gaussian matrix with $p > n$. Suppose $\frac{p}{n} \to \gamma > 1$ as $n, p \to \infty$. Suppose the $\frac{n}{2}$th smallest diagonal element of $\boldsymbol{\Sigma}$ can be lower bounded by a constant $\nu$ with probability $1 - \delta$. Then the minimum eigenvalue of $\frac{1}{n}\bar{\boldsymbol{X}}^\top \bar{\boldsymbol{X}} + \mu\boldsymbol{\Sigma}$ is lower bounded by*

$$\min\left(c_1, c_2\mu\right)$$

*with probability $1 - cn^2 \cdot \exp(-c'n) - \delta$ for some positive constants $c_1, c_2, c, c' > 0$ that only depend on $\gamma$.*

The proof of Lemma 3 is shown in Appendix B.6. Note that

$$\bar{\boldsymbol{S}}_n = \left(\frac{1}{n}\bar{\boldsymbol{X}}_P^\top \bar{\boldsymbol{X}}_P + \mu_n \tilde{\boldsymbol{\Sigma}}_P^{-1}\right)^{-1}$$

where $\bar{\boldsymbol{X}}_P = \tilde{\boldsymbol{X}}_P \tilde{\boldsymbol{\Sigma}}_P^{-1/2}$ is a standard Gaussian matrix. Further, the $\frac{n}{2}$ smallest eigenvalue of $\tilde{\boldsymbol{\Sigma}}_P^{-1}$ is $\frac{n^\kappa}{(2p)^\kappa}$ which converges to a constant $(\frac{\beta}{2\alpha})^\kappa$. Hence, by Lemma 3, we know $\|\bar{\boldsymbol{S}}_n\|_2$ is upper bounded by $O_{\mathrm{p}}(\frac{1}{\mu_n})$ and thus, $\frac{\mu_n}{n}\operatorname{tr}\left(\tilde{\boldsymbol{\Sigma}}_P \tilde{\boldsymbol{S}}_n\right) = O_{\mathrm{p}}(1)$. This completes the proof of Proposition 1. $\qquad\square$

**Proposition 2.**

$$\sup_i |\Delta_i| = O_{\mathrm{p}}\left(\frac{\ln N}{\sqrt{N}}\right).$$

*Proof.* Let us introduce $\bar{\boldsymbol{S}}_n^{\backslash i} = \tilde{\boldsymbol{\Sigma}}_P^{1/2}\boldsymbol{S}_n^{\backslash i}\tilde{\boldsymbol{\Sigma}}_P^{1/2}$ and $\bar{\boldsymbol{x}}_i = \tilde{\boldsymbol{\Sigma}}_P^{-1/2}\tilde{\boldsymbol{x}}_i$. Then,

$$\bar{\boldsymbol{S}}_n^{\backslash i} = \left(\frac{1}{n}\bar{\boldsymbol{X}}_P^\top \bar{\boldsymbol{X}}_P - \frac{1}{n}\bar{\boldsymbol{x}}_i \bar{\boldsymbol{x}}_i^\top + \mu_n \tilde{\boldsymbol{\Sigma}}_P^{-1}\right)^{-1}, \tag{47}$$

where $\bar{\boldsymbol{x}}_i$ is the $i$th row of $\bar{\boldsymbol{X}}_P$. Further, we have

$$\Delta_i = \frac{\mu_n}{n}\operatorname{tr}\left(\bar{\boldsymbol{S}}_n\right) - \frac{\mu_n}{n}\bar{\boldsymbol{x}}_i^\top \bar{\boldsymbol{S}}_n^{\backslash i}\bar{\boldsymbol{x}}_i - \mu_n.$$

To bound $|\Delta_i|$, we can decompose $\Delta_i$ into three parts:

$$\Delta_i = \left(\frac{\mu_n}{n}\operatorname{tr}\left(\bar{\boldsymbol{S}}_n\right) - \frac{\mu_n}{n}\operatorname{tr}\left(\bar{\boldsymbol{S}}_n^{\backslash i}\right)\right) + \left(\frac{\mu_n}{n}\operatorname{tr}\left(\bar{\boldsymbol{S}}_n^{\backslash i}\right) - \frac{\mu_n}{n}\bar{\boldsymbol{x}}_i^\top \bar{\boldsymbol{S}}_n^{\backslash i}\bar{\boldsymbol{x}}_i\right) - \mu_n$$

Intuitively, the first part should be small since $\bar{\boldsymbol{S}}_n$ and $\bar{\boldsymbol{S}}_n^{\backslash i}$ only differ at one sample. For the second part, since $\bar{\boldsymbol{x}}_i$ is independent of $\bar{\boldsymbol{S}}_n^{\backslash i}$, the law of large numbers implies that it should be small as well. Finally, we have $\mu_n \to 0$. We now make these arguments rigorous. By Lemma 3 again, we have

$$\max\left(\|\bar{\boldsymbol{S}}_n\|_2, \max_i \|\bar{\boldsymbol{S}}_n^{\backslash i}\|_2\right) \leq O_{\mathrm{p}}\left(\frac{1}{\mu_n}\right). \tag{48}$$

Then, we can show that the difference between $\frac{\mu_n}{n}\operatorname{tr}\left(\bar{\boldsymbol{S}}_n\right)$ and $\frac{\mu_n}{n}\operatorname{tr}\left(\bar{\boldsymbol{S}}_n^{\backslash i}\right)$ is small. Note that, by the Sherman-Morrison formula,

$$\sup_i \left|\frac{\mu_n}{n}\operatorname{tr}\left(\bar{\boldsymbol{S}}_n\right) - \frac{\mu_n}{n}\operatorname{tr}\left(\bar{\boldsymbol{S}}_n^{\backslash i}\right)\right| = \sup_i \left|\frac{\mu_n}{n}\operatorname{tr}\left(\frac{\bar{\boldsymbol{S}}_n^{\backslash i}\bar{\boldsymbol{x}}_i \bar{\boldsymbol{x}}_i^\top \bar{\boldsymbol{S}}_n^{\backslash i}}{n + \bar{\boldsymbol{x}}_i^\top \bar{\boldsymbol{S}}_n^{\backslash i}\bar{\boldsymbol{x}}_i}\right)\right| = \sup_i \frac{1}{n}\frac{\mu_n \bar{\boldsymbol{x}}_i^\top \left(\bar{\boldsymbol{S}}_n^{\backslash i}\right)^2 \bar{\boldsymbol{x}}_i}{n + \bar{\boldsymbol{x}}_i^\top \bar{\boldsymbol{S}}_n^{\backslash i}\bar{\boldsymbol{x}}_i}$$

$$< \sup_i \frac{1}{n}\frac{\mu_n \bar{\boldsymbol{x}}_i^\top \left(\bar{\boldsymbol{S}}_n^{\backslash i}\right)^2 \bar{\boldsymbol{x}}_i}{\bar{\boldsymbol{x}}_i^\top \bar{\boldsymbol{S}}_n^{\backslash i}\bar{\boldsymbol{x}}_i} \leq \sup_i \frac{\mu_n}{n}\cdot O_{\mathrm{p}}\left(\frac{1}{\mu_n}\right)\cdot \frac{\bar{\boldsymbol{x}}_i^\top \bar{\boldsymbol{S}}_n^{\backslash i}\bar{\boldsymbol{x}}_i}{\bar{\boldsymbol{x}}_i^\top \bar{\boldsymbol{S}}_n^{\backslash i}\bar{\boldsymbol{x}}_i} = O_{\mathrm{p}}\left(\frac{1}{n}\right).$$

Then we want to show the difference between $\frac{\mu_n}{n}\operatorname{tr}\left(\bar{\boldsymbol{S}}_n^{\backslash i}\right)$ and $\frac{\mu_n}{n}\bar{\boldsymbol{x}}_i^\top \bar{\boldsymbol{S}}_n^{\backslash i}\bar{\boldsymbol{x}}_i$ is small. Note that $\bar{\boldsymbol{x}}_i^\top$ is a standard Gaussian vector and it is independent of $\bar{\boldsymbol{S}}_n^{\backslash i}$. Hence, the expectation of $\frac{\mu_n}{n}\bar{\boldsymbol{x}}_i^\top \bar{\boldsymbol{S}}_n^{\backslash i}\bar{\boldsymbol{x}}_i$ is given by $\frac{\mu_n}{n}\operatorname{tr}\left(\bar{\boldsymbol{S}}_n^{\backslash i}\right)$. Further, by standard $\chi^2$ tail bounds [10], we have

$$\mathbb{P}\left(\max_i \left|\frac{\mu_n}{n}\bar{\boldsymbol{x}}_i^\top \bar{\boldsymbol{S}}_n^{\backslash i}\bar{\boldsymbol{x}}_i - \frac{\mu_n}{n}\operatorname{tr}\left(\bar{\boldsymbol{S}}_n^{\backslash i}\right)\right| \geq \frac{2\mu_n p}{n}(\epsilon + \epsilon^2)\|\bar{\boldsymbol{S}}_n^{\backslash i}\|\right) \leq e^{-\epsilon^2 p}. \tag{49}$$

Choose $\epsilon = \frac{\log n}{\sqrt{p}}$, we know

$$\sup_i \left|\frac{\mu_n}{n}\bar{\boldsymbol{x}}_i^\top \bar{\boldsymbol{S}}_n^{\backslash i}\bar{\boldsymbol{x}}_i - \frac{\mu_n}{n}\operatorname{tr}\left(\bar{\boldsymbol{S}}_n^{\backslash i}\right)\right| = O_{\mathrm{p}}\left(\frac{\ln N}{\sqrt{N}}\right). \tag{50}$$

Hence, we have

$$|\Delta_i| \leq \sup_i \left|\frac{\mu_n}{n}\operatorname{tr}\left(\bar{\boldsymbol{S}}_n\right) - \frac{\mu_n}{n}\operatorname{tr}\left(\bar{\boldsymbol{S}}_n^{\backslash i}\right)\right| + \sup_i \left|\frac{\mu_n}{n}\bar{\boldsymbol{x}}_i^\top \bar{\boldsymbol{S}}_n^{\backslash i}\bar{\boldsymbol{x}}_i - \frac{\mu_n}{n}\operatorname{tr}\left(\bar{\boldsymbol{S}}_n^{\backslash i}\right)\right| + |\mu_n| = O_{\mathrm{p}}\left(\frac{\ln N}{\sqrt{N}}\right).$$

$\qquad\square$

## B.3 Analysis of part 2

In this section, we will prove that

$$\text{part 2} \xrightarrow{\text{P}} N^{1-\kappa} \cdot \frac{m'_\kappa(0)}{m^2_\kappa(0)} \cdot \int_\alpha^1 t^{\kappa-2}\,\mathrm{d}t + o_\mathrm{p}(N^{1-\kappa}).$$

We apply a proof similar to that of Theorem 1 in Appendix A. The conditional expectation of part 2 given $\boldsymbol{X}_P$ is

$$\mathbb{E}[\text{part 2} \mid \boldsymbol{X}_P] \;=\; \text{tr}\left(\boldsymbol{\Sigma}_{P^c}\right) \cdot \left(\text{tr}\left(\boldsymbol{\Sigma}_P \boldsymbol{X}_P^\top \left(\boldsymbol{X}_P \boldsymbol{X}_P^\top\right)^{-2} \boldsymbol{X}_P\right) + 1\right). \tag{51}$$

(This expectation only conditions on $\boldsymbol{X}_P$; in particular, it averages over $\boldsymbol{X}_{P^c}$.) The variance of part 2 given $\boldsymbol{X}_P$ is

$$\begin{aligned}
\text{var}\left(\text{part 2} \mid \boldsymbol{X}_P\right) &\leq\; 2 \cdot \text{tr}\left(\boldsymbol{\Sigma}^2_{P^c}\right) \cdot \left\|\left(\boldsymbol{X}_P \boldsymbol{X}_P^\top\right)^{-1} \boldsymbol{X}_P \boldsymbol{\Sigma}_P \boldsymbol{X}_P^\top \left(\boldsymbol{X}_P \boldsymbol{X}_P^\top\right)^{-1}\right\|^2_F \\
&=\; 2 \cdot \text{tr}\left(\boldsymbol{\Sigma}^2_{P^c}\right) \cdot \text{tr}\left(\left(\boldsymbol{\Sigma}_P \boldsymbol{X}_P^\top \left(\boldsymbol{X}_P \boldsymbol{X}_P^\top\right)^{-2} \boldsymbol{X}_P\right)^2\right). \tag{52}
\end{aligned}$$

Let

$$\psi := \text{tr}\left(\boldsymbol{\Sigma}_P \boldsymbol{X}_P^\top \left(\boldsymbol{X}_P \boldsymbol{X}_P^\top\right)^{-2} \boldsymbol{X}_P\right).$$

Then by Markov's inequality, we have

$$\text{part 2} \;=\; \text{tr}\left(\boldsymbol{\Sigma}_{P^c}\right) \cdot (\psi + 1) + O_\mathrm{p}\left(\frac{\sqrt{N-p}}{N^\kappa} \cdot \psi\right). \tag{53}$$

By (29), we have

$$\text{tr}\left(\boldsymbol{\Sigma}_{P^c}\right) \;\to\; N^{1-\kappa} \int_\alpha^1 t^{-\kappa}\,\mathrm{d}t.$$

Hence, we just need to show

$$\psi + 1 \quad \xrightarrow{\text{P}} \quad \frac{m'_\kappa(0)}{m^2_\kappa(0)}, \tag{54}$$

as this will imply

$$\text{part 2} \xrightarrow{\text{P}} N^{1-\kappa} \cdot \frac{m'_\kappa(0)}{m^2_\kappa(0)} \cdot \int_\alpha^1 t^{-\kappa}\,\mathrm{d}t + o_\mathrm{p}(N^{1-\kappa})$$

as required.

To prove (54), let us first rescale $\boldsymbol{\Sigma}$ to $\tilde{\boldsymbol{\Sigma}}$ and introduce the positive sequence $(\mu_n)_{n\geq 1}$ just like what we did for part 1, and with foresight, we pick the sequence such that

$$\mu_n = o(N^{-\kappa}).$$

Then we have

$$\begin{aligned}
\psi + 1 &=\; \text{tr}\left(\tilde{\boldsymbol{\Sigma}}_P \tilde{\boldsymbol{X}}_P^\top \left(\tilde{\boldsymbol{X}}_P \tilde{\boldsymbol{X}}_P^\top\right)^{-2} \tilde{\boldsymbol{X}}_P\right) + 1 \\
&=\; \frac{1}{n} \text{tr}\left(\tilde{\boldsymbol{\Sigma}}_P \left(\frac{1}{n}\tilde{\boldsymbol{X}}_P^\top \tilde{\boldsymbol{X}}_P + \mu_n \boldsymbol{I}\right)^{-1} \left(\frac{1}{n}\tilde{\boldsymbol{X}}_P^\top \tilde{\boldsymbol{X}}_P\right) \left(\frac{1}{n}\tilde{\boldsymbol{X}}_P^\top \tilde{\boldsymbol{X}}_P + \mu_n \boldsymbol{I}\right)^{-1}\right) + \epsilon'_{\mu_n} + 1 \\
&=\; \frac{1}{n} \text{tr}\left(\tilde{\boldsymbol{\Sigma}}_P \tilde{\boldsymbol{S}}_n\right) - \frac{\mu_n}{n} \text{tr}\left(\tilde{\boldsymbol{\Sigma}}_P \tilde{\boldsymbol{S}}^2_n\right) + 1 + \epsilon'_{\mu_n},
\end{aligned}$$

where $\epsilon'_{\mu_n}$ is given by

$$\begin{aligned}
\epsilon'_{\mu_n} &=\; \frac{1}{n} \text{tr}\left(\tilde{\boldsymbol{\Sigma}}_P \frac{1}{\sqrt{n}}\tilde{\boldsymbol{X}}_P^\top \left(\frac{1}{n}\tilde{\boldsymbol{X}}_P \tilde{\boldsymbol{X}}_P^\top\right)^{-2} \frac{1}{\sqrt{n}}\tilde{\boldsymbol{X}}_P\right) \\
&\quad - \frac{1}{n} \text{tr}\left(\tilde{\boldsymbol{\Sigma}}_P \left(\frac{1}{n}\tilde{\boldsymbol{X}}_P^\top \tilde{\boldsymbol{X}}_P + \mu_n \boldsymbol{I}\right)^{-1} \left(\frac{1}{n}\tilde{\boldsymbol{X}}_P^\top \tilde{\boldsymbol{X}}_P\right) \left(\frac{1}{n}\tilde{\boldsymbol{X}}_P^\top \tilde{\boldsymbol{X}}_P + \mu_n \boldsymbol{I}\right)^{-1}\right).
\end{aligned}$$

We shall prove the following:

$$|\epsilon'_{\mu_n}| \;=\; o_{\mathrm p}(1), \tag{55}$$

and

$$\frac{1}{n}\operatorname{tr}\left(\tilde{\boldsymbol\Sigma}_P \tilde{\boldsymbol S}_n\right) - \frac{\mu_n}{n}\operatorname{tr}\left(\tilde{\boldsymbol\Sigma}_P \tilde{\boldsymbol S}_n^2\right) + 1 \;\xrightarrow{\ \mathrm P\ }\; \frac{m'_\kappa(0)}{m_\kappa^2(0)}, \tag{56}$$

which suffices to establish (54).

### B.3.1 Proof of Equation (55)

To bound $|\epsilon'_{\mu_n}|$, note that

$$
|\epsilon'_{\mu_n}| \;\overset{\text{(i)}}{\leq}\; \frac{1}{n}\|\tilde{\boldsymbol\Sigma}_P\|_2 \left( \operatorname{tr}\left( \frac{1}{\sqrt{n}}\tilde{\boldsymbol X}_P^\top \left(\frac{1}{n}\tilde{\boldsymbol X}_P \tilde{\boldsymbol X}_P^\top\right)^{-2} \frac{1}{\sqrt{n}}\tilde{\boldsymbol X}_P \right) \right.
$$
$$
\left. - \operatorname{tr}\left( \left(\frac{1}{n}\tilde{\boldsymbol X}_P^\top \tilde{\boldsymbol X}_P + \mu_n \boldsymbol I\right)^{-1} \left(\frac{1}{n}\tilde{\boldsymbol X}_P^\top \tilde{\boldsymbol X}_P\right) \left(\frac{1}{n}\tilde{\boldsymbol X}_P^\top \tilde{\boldsymbol X}_P + \mu_n \boldsymbol I\right)^{-1} \right) \right)
$$
$$
\leq\; \frac{N^\kappa}{n}\cdot \sum_{i=1}^{n} \frac{\mu_n(2\tilde\lambda_i + \mu_n)}{(\tilde\lambda_i + \mu_n)^2 \tilde\lambda_i} \;\leq\; 2N^\kappa \cdot \frac{\mu_n}{\min_i(\tilde\lambda_i^2)}, \tag{57}
$$

where $\tilde\lambda_i$ is the $i$-th eigenvalue of $\frac{1}{n}\tilde{\boldsymbol X}_P \tilde{\boldsymbol X}_P^\top$ and inequality (i) holds due to the fact that

$$
\frac{1}{\sqrt{n}}\tilde{\boldsymbol X}_P^\top \left(\frac{1}{n}\tilde{\boldsymbol X}_P \tilde{\boldsymbol X}_P^\top\right)^{-2}\frac{1}{\sqrt{n}}\tilde{\boldsymbol X}_P - \left(\frac{1}{n}\tilde{\boldsymbol X}_P^\top \tilde{\boldsymbol X}_P + \mu_n \boldsymbol I\right)^{-1}\left(\frac{1}{n}\tilde{\boldsymbol X}_P^\top \tilde{\boldsymbol X}_P\right)\left(\frac{1}{n}\tilde{\boldsymbol X}_P^\top \tilde{\boldsymbol X}_P + \mu_n \boldsymbol I\right)^{-1}
$$

is positive semi-definite. By Lemma 1, Lemma 2, and (57), since $\mu_n = o(N^{-\kappa})$, we have

$$|\epsilon'_{\mu_n}| \;=\; o_{\mathrm p}(1).$$

### B.3.2 Proof of Equation (56)

We now prove

$$\frac{1}{n}\operatorname{tr}\left(\tilde{\boldsymbol\Sigma}_P \tilde{\boldsymbol S}_n\right) - \frac{\mu_n}{n}\operatorname{tr}\left(\tilde{\boldsymbol\Sigma}_P \tilde{\boldsymbol S}_n^2\right) + 1 \;\xrightarrow{\ \mathrm P\ }\; \frac{m'_\kappa(0)}{m_\kappa^2(0)}.$$

Towards this goal, we employ a strategy similar to the proof of (38). Using the identity $\tilde{\boldsymbol S}_n^{-1} - \mu_n \boldsymbol I = \frac{1}{n}\sum_{i=1}^n \tilde{\boldsymbol x}_i \tilde{\boldsymbol x}_i^\top$, we have

$$
\frac{1}{n}\sum_{i=1}^{n}\tilde{\boldsymbol x}_i^\top \tilde{\boldsymbol S}_n^2 \tilde{\boldsymbol x}_i \;=\; \frac{1}{n}\operatorname{tr}\left(\sum_{i=1}^{n}\tilde{\boldsymbol S}_n^2 \tilde{\boldsymbol x}_i \tilde{\boldsymbol x}_i^\top\right)
$$
$$
=\; \operatorname{tr}\left(\tilde{\boldsymbol S}_n^2 (\tilde{\boldsymbol S}_n^{-1} - \mu_n \boldsymbol I)\right)
$$
$$
=\; \operatorname{tr}\left(\tilde{\boldsymbol S}_n - \mu_n \tilde{\boldsymbol S}_n^2\right).
$$

With (44), we have

$$
\operatorname{tr}\left(\tilde{\boldsymbol S}_n - \mu_n \tilde{\boldsymbol S}_n^2\right) \;=\; \frac{1}{n}\sum_{i=1}^{n}\tilde{\boldsymbol x}_i^\top \tilde{\boldsymbol S}_n^2 \tilde{\boldsymbol x}_i
$$
$$
=\; \frac{1}{n}\sum_{i=1}^{n}\tilde{\boldsymbol x}_i^\top \left(\tilde{\boldsymbol S}_n^{\backslash i} - \frac{1}{n}\cdot \frac{\tilde{\boldsymbol S}_n^{\backslash i}\tilde{\boldsymbol x}_i \tilde{\boldsymbol x}_i^\top \tilde{\boldsymbol S}_n^{\backslash i}}{1 + \frac{1}{n}\tilde{\boldsymbol x}_i^\top \tilde{\boldsymbol S}_n^{\backslash i}\tilde{\boldsymbol x}_i}\right)^2 \tilde{\boldsymbol x}_i
$$
$$
=\; \sum_{i=1}^{n}\frac{\frac{1}{n}\tilde{\boldsymbol x}_i^\top \left(\tilde{\boldsymbol S}_n^{\backslash i}\right)^2 \tilde{\boldsymbol x}_i}{\left(1 + \frac{1}{n}\tilde{\boldsymbol x}_i^\top \tilde{\boldsymbol S}_n^{\backslash i}\tilde{\boldsymbol x}_i\right)^2}. \tag{58}
$$

451 Note that $\frac{1}{n}\operatorname{tr}\left(\tilde{\boldsymbol{S}}_n - \mu_n\tilde{\boldsymbol{S}}_n^2\right) = m_n(-\mu_n) - \mu_n m_n'(-\mu_n)$. With (45) and (58), we have

$$
\begin{aligned}
-\mu_n m_n'(-\mu_n) &= \frac{1}{n}\sum_{i=1}^n \frac{\frac{1}{n}\tilde{\boldsymbol{x}}_i^\top\left(\tilde{\boldsymbol{S}}_n^{\backslash i}\right)^2\tilde{\boldsymbol{x}}_i}{\left(1+\frac{1}{n}\tilde{\boldsymbol{x}}_i^\top\tilde{\boldsymbol{S}}_n^{\backslash i}\tilde{\boldsymbol{x}}_i\right)^2} - m_n(-\mu_n) \\[2mm]
&= \frac{1}{n}\sum_{i=1}^n \frac{\frac{1}{n}\tilde{\boldsymbol{x}}_i^\top\left(\tilde{\boldsymbol{S}}_n^{\backslash i}\right)^2\tilde{\boldsymbol{x}}_i}{\left(1+\frac{1}{n}\tilde{\boldsymbol{x}}_i^\top\tilde{\boldsymbol{S}}_n^{\backslash i}\tilde{\boldsymbol{x}}_i\right)^2} - \frac{1}{n}\sum_{i=1}^n \frac{1}{\mu_n + \frac{\mu_n}{n}\tilde{\boldsymbol{x}}_i^\top\tilde{\boldsymbol{S}}_n^{\backslash i}\tilde{\boldsymbol{x}}_i} \\[2mm]
&= \mu_n\cdot\frac{1}{n}\sum_{i=1}^n \frac{\frac{\mu_n}{n}\tilde{\boldsymbol{x}}_i^\top\left(\tilde{\boldsymbol{S}}_n^{\backslash i}\right)^2\tilde{\boldsymbol{x}}_i - 1 - \frac{1}{n}\tilde{\boldsymbol{x}}_i^\top\tilde{\boldsymbol{S}}_n^{\backslash i}\tilde{\boldsymbol{x}}_i}{\left(\mu_n + \frac{\mu_n}{n}\tilde{\boldsymbol{x}}_i^\top\tilde{\boldsymbol{S}}_n^{\backslash i}\tilde{\boldsymbol{x}}_i\right)^2}.
\end{aligned}
$$

452 Hence, we have

$$
m_n'(-\mu_n) = \frac{1}{n}\sum_{i=1}^n \frac{1+\frac{1}{n}\tilde{\boldsymbol{x}}_i^\top\left(\tilde{\boldsymbol{S}}_n^{\backslash i} - \mu_n\left(\tilde{\boldsymbol{S}}_n^{\backslash i}\right)^2\right)\tilde{\boldsymbol{x}}_i}{\left(\mu_n + \frac{\mu_n}{n}\tilde{\boldsymbol{x}}_i^\top\tilde{\boldsymbol{S}}_n^{\backslash i}\tilde{\boldsymbol{x}}_i\right)^2}. \tag{59}
$$

453 Note that

$$
|m_n'(-\mu_n) - m_n'(0)| \le \frac{2\mu_n}{\min(\tilde{\lambda}_i^3)},
$$

454 where $\tilde{\lambda}_1,\ldots,\tilde{\lambda}_n$ are the eigenvalues of $\frac{1}{n}\tilde{\boldsymbol{X}}_P\tilde{\boldsymbol{X}}_P^\top$. Therefore, by Lemma 1, we have

$$
m_n'(-\mu_n) = m_n'(0) + O_{\mathrm{p}}(\mu_n) \xrightarrow{\mathrm{P}} m_\kappa'(0).
$$

455 From (38), Proposition 1, and Proposition 2, we know that

$$
\left(\mu_n + \frac{\mu_n}{n}\tilde{\boldsymbol{x}}_i^\top\tilde{\boldsymbol{S}}_n^{\backslash i}\tilde{\boldsymbol{x}}_i\right)^2 \xrightarrow{\mathrm{P}} \frac{1}{m_\kappa^2(0)} > 0. \tag{60}
$$

456 We claim that

$$
\frac{1}{n}\operatorname{tr}\left(\tilde{\boldsymbol{\Sigma}}_P\tilde{\boldsymbol{S}}_n - \mu_n\tilde{\boldsymbol{\Sigma}}_P\tilde{\boldsymbol{S}}_n^2\right) = O_{\mathrm{p}}(1), \tag{61}
$$

$$
\frac{1}{n}\tilde{\boldsymbol{x}}_i^\top\left(\tilde{\boldsymbol{S}}_n^{\backslash i} - \mu_n\left(\tilde{\boldsymbol{S}}_n^{\backslash i}\right)^2\right)\tilde{\boldsymbol{x}}_i = \frac{1}{n}\operatorname{tr}\left(\tilde{\boldsymbol{\Sigma}}_P\tilde{\boldsymbol{S}}_n - \mu_n\tilde{\boldsymbol{\Sigma}}_P\tilde{\boldsymbol{S}}_n^2\right) + O_{\mathrm{p}}\left(\frac{\ln N}{\sqrt{N}}\right) \tag{62}
$$

457 (Proposition 3 and Proposition 4 below). So, we obtain from (59)

$$
m_n'(-\mu_n) = \frac{1}{n}\sum_{i=1}^n \frac{1+\frac{1}{n}\tilde{\boldsymbol{x}}_i^\top\left(\tilde{\boldsymbol{S}}_n^{\backslash i} - \mu_n\left(\tilde{\boldsymbol{S}}_n^{\backslash i}\right)^2\right)\tilde{\boldsymbol{x}}_i}{\left(\mu_n + \frac{\mu_n}{n}\tilde{\boldsymbol{x}}_i^\top\tilde{\boldsymbol{S}}_n^{\backslash i}\tilde{\boldsymbol{x}}_i\right)^2} \xrightarrow{\mathrm{P}} \frac{1+\frac{1}{n}\operatorname{tr}\left(\tilde{\boldsymbol{\Sigma}}_P\tilde{\boldsymbol{S}}_n - \mu_n\tilde{\boldsymbol{\Sigma}}_P\tilde{\boldsymbol{S}}_n^2\right)}{1/m_\kappa(0)^2},
$$

458 i.e.,

$$
\frac{m_n'(-\mu_n)}{m_\kappa(0)^2} \xrightarrow{\mathrm{P}} 1 + \frac{1}{n}\operatorname{tr}\left(\tilde{\boldsymbol{\Sigma}}_P\tilde{\boldsymbol{S}}_n - \mu_n\tilde{\boldsymbol{\Sigma}}_P\tilde{\boldsymbol{S}}_n^2\right).
$$

459 This suffices to prove (56) as required.

### B.3.3  Supporting propositions

**Proposition 3.**

$$\frac{1}{n}\operatorname{tr}\left(\tilde{\boldsymbol{\Sigma}}_P\tilde{\boldsymbol{S}}_n - \mu_n\tilde{\boldsymbol{\Sigma}}_P\tilde{\boldsymbol{S}}_n^2\right) \;=\; O_{\mathrm p}(1).$$

*Proof.* Recall that

$$\bar{\boldsymbol{S}}_n \;=\; \tilde{\boldsymbol{\Sigma}}_P^{1/2}\tilde{\boldsymbol{S}}_n\tilde{\boldsymbol{\Sigma}}_P^{1/2} \;=\; \left(\frac{1}{n}\bar{\boldsymbol{X}}_P^\top\bar{\boldsymbol{X}}_P + \mu_n\tilde{\boldsymbol{\Sigma}}_P^{-1}\right)^{-1},$$

where $\bar{\boldsymbol{X}}_P = \tilde{\boldsymbol{X}}_P\tilde{\boldsymbol{\Sigma}}_P^{-1/2}$ is a standard Gaussian matrix. Let $\frac{1}{n}\bar{\boldsymbol{X}}_P^\top\bar{\boldsymbol{X}}_P = \boldsymbol{U}\boldsymbol{\Lambda}\boldsymbol{U}^\top$ be the singular value decomposition of $\frac{1}{n}\bar{\boldsymbol{X}}_P^\top\bar{\boldsymbol{X}}_P$, where $\boldsymbol{U}\boldsymbol{U}^\top = \boldsymbol{I}$ and $\boldsymbol{\Lambda}$ is a diagonal matrix with

$$\Lambda_{1,1} \geq \Lambda_{2,2} \geq \cdots \geq \Lambda_{n,n} \geq \Lambda_{n+1,n+1} = \cdots = \Lambda_{p,p} = 0.$$

Hence, we have

$$\tilde{\boldsymbol{\Sigma}}_P^{1/2}\tilde{\boldsymbol{S}}_n\tilde{\boldsymbol{\Sigma}}_P^{1/2} - \mu_n\tilde{\boldsymbol{\Sigma}}_P^{1/2}\tilde{\boldsymbol{S}}_n^2\tilde{\boldsymbol{\Sigma}}_P^{1/2} \;=\; \bar{\boldsymbol{S}}_n\left(\frac{1}{n}\bar{\boldsymbol{X}}_P^\top\bar{\boldsymbol{X}}_P\right)\bar{\boldsymbol{S}}_n$$
$$=\; \left(\boldsymbol{\Lambda}+\mu_n\boldsymbol{U}^\top\tilde{\boldsymbol{\Sigma}}_P^{-1}\boldsymbol{U}\right)^{-1}\boldsymbol{\Lambda}\left(\boldsymbol{\Lambda}+\mu_n\boldsymbol{U}^\top\tilde{\boldsymbol{\Sigma}}_P^{-1}\boldsymbol{U}\right)^{-1}.$$

Our next step is to bound the maximum eigenvalue of

$$\left(\boldsymbol{\Lambda}+\mu_n\boldsymbol{U}^\top\tilde{\boldsymbol{\Sigma}}_P^{-1}\boldsymbol{U}\right)^{-1}\boldsymbol{\Lambda}\left(\boldsymbol{\Lambda}+\mu_n\boldsymbol{U}^\top\tilde{\boldsymbol{\Sigma}}_P^{-1}\boldsymbol{U}\right)^{-1}.$$

Let $\phi_n$ be the smallest eigenvalue of $\tilde{\boldsymbol{\Sigma}}_P^{-1}$. Define $\boldsymbol{\Lambda}_\phi = \boldsymbol{\Lambda} + \frac{\mu_n\phi_n}{2}\boldsymbol{I}$ and $\boldsymbol{\Sigma}_\phi^{-1} = \mu_n(\tilde{\boldsymbol{\Sigma}}_P^{-1} - \frac{\phi_n}{2}\boldsymbol{I})$. Then $\boldsymbol{\Lambda}_\phi$ and $\boldsymbol{\Sigma}_\phi$ are two positive definite diagonal matrices. Intuitively, for $\mu_n$ small enough,

$$\left(\boldsymbol{\Lambda}+\mu_n\boldsymbol{U}^\top\tilde{\boldsymbol{\Sigma}}_P^{-1}\boldsymbol{U}\right)^{-1}\boldsymbol{\Lambda}\left(\boldsymbol{\Lambda}+\mu_n\boldsymbol{U}^\top\tilde{\boldsymbol{\Sigma}}_P^{-1}\boldsymbol{U}\right)^{-1} \approx \boldsymbol{\Lambda}_\phi^{-1}\boldsymbol{\Lambda}\boldsymbol{\Lambda}_\phi^{-1},$$

the latter having a maximum eigenvalue bounded by a constant. We now make this argument rigorous. By the Sherman-Morrision formula, we have

$$\left(\boldsymbol{\Lambda}+\mu_n\boldsymbol{U}^\top\tilde{\boldsymbol{\Sigma}}_P^{-1}\boldsymbol{U}\right)^{-1} \;=\; \left(\boldsymbol{\Lambda}_\phi+\boldsymbol{U}^\top\boldsymbol{\Sigma}_\phi^{-1}\boldsymbol{U}\right)^{-1}$$
$$=\; \boldsymbol{\Lambda}_\phi^{-1} - \boldsymbol{\Lambda}_\phi^{-1}\boldsymbol{U}^\top\left(\boldsymbol{\Sigma}_\phi+\boldsymbol{U}^\top\boldsymbol{\Lambda}_\phi^{-1}\boldsymbol{U}\right)^{-1}\boldsymbol{U}\boldsymbol{\Lambda}_\phi^{-1}.$$

Hence, we know

$$\left\|\left(\boldsymbol{\Lambda}+\mu_n\boldsymbol{U}^\top\tilde{\boldsymbol{\Sigma}}_P^{-1}\boldsymbol{U}\right)^{-1}\boldsymbol{\Lambda}\left(\boldsymbol{\Lambda}+\mu_n\boldsymbol{U}^\top\tilde{\boldsymbol{\Sigma}}_P^{-1}\boldsymbol{U}\right)^{-1}\right\|_2$$
$$\leq\; 2\left\|\boldsymbol{\Lambda}_\phi^{-1}\boldsymbol{\Lambda}\boldsymbol{\Lambda}_\phi^{-1}\right\|_2$$
$$+2\left\|\boldsymbol{\Lambda}_\phi^{-1}\boldsymbol{U}^\top\left(\boldsymbol{\Sigma}_\phi+\boldsymbol{U}^\top\boldsymbol{\Lambda}_\phi^{-1}\boldsymbol{U}\right)^{-1}\boldsymbol{U}\boldsymbol{\Lambda}_\phi^{-1}\boldsymbol{\Lambda}\boldsymbol{\Lambda}_\phi^{-1}\boldsymbol{U}^\top\left(\boldsymbol{\Sigma}_\phi+\boldsymbol{U}^\top\boldsymbol{\Lambda}_\phi^{-1}\boldsymbol{U}\right)^{-1}\boldsymbol{U}\boldsymbol{\Lambda}_\phi^{-1}\right\|_2$$
$$\leq\; 2\left\|\boldsymbol{\Lambda}_\phi^{-1}\boldsymbol{\Lambda}\boldsymbol{\Lambda}_\phi^{-1}\right\|_2\left(1+\left\|\boldsymbol{\Lambda}_\phi^{-1}\boldsymbol{U}^\top\left(\boldsymbol{\Sigma}_\phi+\boldsymbol{U}^\top\boldsymbol{\Lambda}_\phi^{-1}\boldsymbol{U}\right)^{-1}\boldsymbol{U}\boldsymbol{U}^\top\left(\boldsymbol{\Sigma}_\phi+\boldsymbol{U}^\top\boldsymbol{\Lambda}_\phi^{-1}\boldsymbol{U}\right)^{-1}\boldsymbol{U}\boldsymbol{\Lambda}_\phi^{-1}\right\|_2\right)$$
$$=\; 2\left\|\boldsymbol{\Lambda}_\phi^{-1}\boldsymbol{\Lambda}\boldsymbol{\Lambda}_\phi^{-1}\right\|_2\left(1+\left\|\boldsymbol{\Lambda}_\phi^{-1}\left(\boldsymbol{U}\boldsymbol{\Sigma}_\phi\boldsymbol{U}^\top+\boldsymbol{\Lambda}_\phi^{-1}\right)^{-2}\boldsymbol{\Lambda}_\phi^{-1}\right\|_2\right)$$
$$=\; 2\left\|\boldsymbol{\Lambda}_\phi^{-1}\boldsymbol{\Lambda}\boldsymbol{\Lambda}_\phi^{-1}\right\|_2\left(1+\left\|\left(\boldsymbol{\Lambda}_\phi\left(\boldsymbol{U}\boldsymbol{\Sigma}_\phi\boldsymbol{U}^\top+\boldsymbol{\Lambda}_\phi^{-1}\right)^2\boldsymbol{\Lambda}_\phi\right)^{-1}\right\|_2\right)$$
$$=\; 2\left\|\boldsymbol{\Lambda}_\phi^{-1}\boldsymbol{\Lambda}\boldsymbol{\Lambda}_\phi^{-1}\right\|_2\left(1+\left\|\left(\boldsymbol{I}+\boldsymbol{\Lambda}_\phi\left(\boldsymbol{U}\boldsymbol{\Sigma}_\phi\boldsymbol{U}^\top\boldsymbol{\Lambda}_\phi^{-1}+\boldsymbol{\Lambda}_\phi^{-1}\boldsymbol{U}\boldsymbol{\Sigma}_\phi\boldsymbol{U}^\top+(\boldsymbol{U}\boldsymbol{\Sigma}_\phi\boldsymbol{U}^\top)^2\right)\boldsymbol{\Lambda}_\phi\right)^{-1}\right\|_2\right)$$
$$\leq\; 4\left\|\boldsymbol{\Lambda}_\phi^{-1}\boldsymbol{\Lambda}\boldsymbol{\Lambda}_\phi^{-1}\right\|_2.$$

471 Note that $\mathbf{\Lambda}_\phi$ and $\mathbf{\Lambda}$ are both diagonal. Hence, we have

$$\|\mathbf{\Lambda}_\phi^{-1}\mathbf{\Lambda}\mathbf{\Lambda}_\phi^{-1}\|_2 \;=\; \max_{1\le i\le n}\frac{\Lambda_{i,i}}{\left(\Lambda_{i,i}+\frac{\phi_n\mu_n}{2}\right)^2} \;\le\; \frac{1}{\Lambda_{n,n}}.$$

472 To lower bound $\Lambda_{n,n}$, we use the following lemma.

473 **Lemma 4** (Lemma 10 of 19). *Let $\mathbf{X}\in\mathbb{R}^{n\times p}$ be a standard Gaussian random matrix, and let $\mathbf{x}_i$*
474 *be the $i$-th row of the matrix $\mathbf{X}$. Let $\rho = n/p > 1$. There exist constants $c, c' > 0$ such that for*
475 *large enough $n$, with probability at least $1 - c'(p^2 + n^2)e^{-cn}$, the eigenvalues of $\frac{1}{n}\mathbf{X}^\top\mathbf{X}$ and of*
476 $\frac{1}{n}(\mathbf{X}^\top\mathbf{X} - \mathbf{x}_i\mathbf{x}_i^\top)$ *for each $i = 1,\dots,n$ are contained in the interval*

$$\left(\frac{1}{2}\cdot\min\left\{(1-1/\sqrt{\rho})^2, 1/\rho\right\}, 9\rho^2\right).$$

477 Hence, by Lemma 4, we have $\Lambda_{n,n} \ge \frac{1}{2}\min((1-\sqrt{\beta/\alpha})^2, \beta/\alpha) > 0$ hold with probability
478 $1 - c\cdot n^2\exp(-c'n)$ for some absolute constants $c, c' > 0$. Hence, we have

$$\left\|\bar{\mathbf{S}}_n\left(\frac{1}{n}\bar{\mathbf{X}}_P^\top\bar{\mathbf{X}}_P\right)\bar{\mathbf{S}}_n\right\|_2 \;\le\; O_{\mathrm{p}}(1) \tag{63}$$

479 as required. $\qquad\qquad\qquad\qquad\qquad\qquad\qquad\qquad\qquad\qquad\qquad\qquad\qquad\qquad$ $\square$

**Proposition 4.**

$$\frac{1}{n}\tilde{\mathbf{x}}_i^\top\left(\tilde{\mathbf{S}}_n^{\backslash i} - \mu_n\left(\tilde{\mathbf{S}}_n^{\backslash i}\right)^2\right)\tilde{\mathbf{x}}_i \;=\; \frac{1}{n}\operatorname{tr}\left(\tilde{\mathbf{\Sigma}}_P\tilde{\mathbf{S}}_n - \mu_n\tilde{\mathbf{\Sigma}}_P\tilde{\mathbf{S}}_n^2\right) + O_{\mathrm{p}}\left(\frac{\ln N}{\sqrt{N}}\right).$$

480 *Proof.* It is clear that we just need to prove the following two arguments

$$\sup_i\left|\frac{1}{n}\tilde{\mathbf{x}}_i^\top\left(\tilde{\mathbf{S}}_n^{\backslash i} - \mu_n\left(\tilde{\mathbf{S}}_n^{\backslash i}\right)^2\right)\tilde{\mathbf{x}}_i - \frac{1}{n}\operatorname{tr}\left(\tilde{\mathbf{\Sigma}}_P\tilde{\mathbf{S}}_n^{\backslash i} - \mu_n\tilde{\mathbf{\Sigma}}_P\left(\tilde{\mathbf{S}}_n^{\backslash i}\right)^2\right)\right| \;=\; O_{\mathrm{p}}\left(\frac{\ln N}{\sqrt{N}}\right)$$

$$\tag{64}$$

$$\sup_i\left|\frac{1}{n}\operatorname{tr}\left(\tilde{\mathbf{\Sigma}}_P\tilde{\mathbf{S}}_n - \mu_n\tilde{\mathbf{\Sigma}}_P\left(\tilde{\mathbf{S}}_n\right)^2\right) - \frac{1}{n}\operatorname{tr}\left(\tilde{\mathbf{\Sigma}}_P\tilde{\mathbf{S}}_n^{\backslash i} - \mu_n\tilde{\mathbf{\Sigma}}_P\left(\tilde{\mathbf{S}}_n^{\backslash i}\right)^2\right)\right| \;=\; O_{\mathrm{p}}\left(\frac{\ln N}{\sqrt{N}}\right).$$

$$\tag{65}$$

481 To show (64), we use a proof similar to that of (63). By Lemma 4, we know

$$\max_i\left\|\bar{\mathbf{S}}_n^{\backslash i}\left(\frac{1}{n}\bar{\mathbf{X}}_P^\top\bar{\mathbf{X}}_P - \frac{1}{n}\bar{\mathbf{x}}_i\bar{\mathbf{x}}_i^\top\right)\bar{\mathbf{S}}_n^{\backslash i}\right\|_2 \;=\; O_{\mathrm{p}}(1). \tag{66}$$

482 Note that

$$\tilde{\mathbf{x}}_i^\top\left(\tilde{\mathbf{S}}_n^{\backslash i} - \mu_n\left(\tilde{\mathbf{S}}_n^{\backslash i}\right)^2\right)\tilde{\mathbf{x}}_i \;=\; \bar{\mathbf{x}}_i^\top\tilde{\mathbf{\Sigma}}_P^{1/2}\left(\tilde{\mathbf{S}}_n^{\backslash i} - \mu_n\left(\tilde{\mathbf{S}}_n^{\backslash i}\right)^2\right)\tilde{\mathbf{\Sigma}}_P^{1/2}\bar{\mathbf{x}}_i$$

$$=\; \bar{\mathbf{x}}_i^\top\left(\bar{\mathbf{S}}_n^{\backslash i}\left(\frac{1}{n}\bar{\mathbf{X}}_P^\top\bar{\mathbf{X}}_P - \frac{1}{n}\bar{\mathbf{x}}_i\bar{\mathbf{x}}_i^\top\right)\bar{\mathbf{S}}_n^{\backslash i}\right)\bar{\mathbf{x}}_i.$$

483 Furthermore, $\tilde{\mathbf{x}}_i$ is a standard Gaussian vector, and it is independent of the matrix

$$\bar{\mathbf{S}}_n^{\backslash i}\left(\frac{1}{n}\bar{\mathbf{X}}_P^\top\bar{\mathbf{X}}_P - \frac{1}{n}\bar{\mathbf{x}}_i\bar{\mathbf{x}}_i^\top\right)\bar{\mathbf{S}}_n^{\backslash i}.$$

Hence, we apply the same proof of (50) with (66) and Lemma 1 of [10]; this gives

$$
\sup_i \left| \frac{1}{n} \bar{\boldsymbol{x}}_i^\top \tilde{\boldsymbol{\Sigma}}_P^{1/2} \left( \tilde{\boldsymbol{S}}_n^{\backslash i} - \mu_n \left( \tilde{\boldsymbol{S}}_n^{\backslash i} \right)^2 \right) \tilde{\boldsymbol{\Sigma}}_P^{1/2} \bar{\boldsymbol{x}}_i - \frac{1}{n} \operatorname{tr} \left( \tilde{\boldsymbol{\Sigma}}_P^{1/2} \left( \tilde{\boldsymbol{S}}_n^{\backslash i} - \mu_n \left( \tilde{\boldsymbol{S}}_n^{\backslash i} \right)^2 \right) \tilde{\boldsymbol{\Sigma}}_P^{1/2} \right) \right|
$$

$$
= O_{\mathrm{p}} \left( \frac{\ln N}{\sqrt{N}} \right).
$$

Hence, (64) holds. Therefore, it remains to show (65), which is equivalent to

$$
\sup_i \left| \frac{1}{n} \operatorname{tr} \left( \bar{\boldsymbol{S}}_n \left( \frac{1}{n} \bar{\boldsymbol{X}}_P^\top \bar{\boldsymbol{X}}_P \right) \bar{\boldsymbol{S}}_n \right) - \frac{1}{n} \operatorname{tr} \left( \bar{\boldsymbol{S}}_n^{\backslash i} \left( \frac{1}{n} \bar{\boldsymbol{X}}_P^\top \bar{\boldsymbol{X}}_P - \frac{1}{n} \bar{\boldsymbol{x}}_i \bar{\boldsymbol{x}}_i^\top \right) \bar{\boldsymbol{S}}_n^{\backslash i} \right) \right| = O_{\mathrm{p}} \left( \frac{\ln N}{\sqrt{N}} \right).
$$
(67)

By the Sherman-Morrison formula, we have

$$
\bar{\boldsymbol{S}}_n = \bar{\boldsymbol{S}}_n^{\backslash i} - \frac{\bar{\boldsymbol{S}}_n^{\backslash i} \bar{\boldsymbol{x}}_i \bar{\boldsymbol{x}}_i^\top \bar{\boldsymbol{S}}_n^{\backslash i}}{n + \bar{\boldsymbol{x}}_i^\top \bar{\boldsymbol{S}}_n^{\backslash i} \bar{\boldsymbol{x}}_i},
$$

and therefore

$$
\operatorname{tr} \left( \bar{\boldsymbol{S}}_n \left( \frac{1}{n} \bar{\boldsymbol{X}}_P^\top \bar{\boldsymbol{X}}_P \right) \bar{\boldsymbol{S}}_n - \bar{\boldsymbol{S}}_n^{\backslash i} \left( \frac{1}{n} \bar{\boldsymbol{X}}_P^\top \bar{\boldsymbol{X}}_P - \frac{1}{n} \bar{\boldsymbol{x}}_i \bar{\boldsymbol{x}}_i^\top \right) \bar{\boldsymbol{S}}_n^{\backslash i} \right)
$$

$$
= \operatorname{tr} \left( \bar{\boldsymbol{S}}_n^{\backslash i} \left( \frac{1}{n} \bar{\boldsymbol{x}}_i \bar{\boldsymbol{x}}_i^\top \right) \bar{\boldsymbol{S}}_n^{\backslash i} \right) - 2 \cdot \operatorname{tr} \left( \bar{\boldsymbol{S}}_n^{\backslash i} \left( \frac{1}{n} \bar{\boldsymbol{X}}_P^\top \bar{\boldsymbol{X}}_P \right) \frac{\bar{\boldsymbol{S}}_n^{\backslash i} \bar{\boldsymbol{x}}_i \bar{\boldsymbol{x}}_i^\top \bar{\boldsymbol{S}}_n^{\backslash i}}{n + \bar{\boldsymbol{x}}_i^\top \bar{\boldsymbol{S}}_n^{\backslash i} \bar{\boldsymbol{x}}_i} \right)
$$

$$
+ \operatorname{tr} \left( \frac{\bar{\boldsymbol{S}}_n^{\backslash i} \bar{\boldsymbol{x}}_i \bar{\boldsymbol{x}}_i^\top \bar{\boldsymbol{S}}_n^{\backslash i}}{n + \bar{\boldsymbol{x}}_i^\top \bar{\boldsymbol{S}}_n^{\backslash i} \bar{\boldsymbol{x}}_i} \left( \frac{1}{n} \bar{\boldsymbol{X}}_P^\top \bar{\boldsymbol{X}}_P \right) \frac{\bar{\boldsymbol{S}}_n^{\backslash i} \bar{\boldsymbol{x}}_i \bar{\boldsymbol{x}}_i^\top \bar{\boldsymbol{S}}_n^{\backslash i}}{n + \bar{\boldsymbol{x}}_i^\top \bar{\boldsymbol{S}}_n^{\backslash i} \bar{\boldsymbol{x}}_i} \right).
$$
(68)

Let $\boldsymbol{M}_i = \bar{\boldsymbol{S}}_n^{\backslash i} \left( \frac{1}{n} \bar{\boldsymbol{X}}_P^\top \bar{\boldsymbol{X}}_P - \frac{1}{n} \bar{\boldsymbol{x}}_i \bar{\boldsymbol{x}}_i^\top \right) \bar{\boldsymbol{S}}_n^{\backslash i}$. Let $\rho = \frac{\mu_n}{n} \bar{\boldsymbol{x}}_i^\top \bar{\boldsymbol{S}}_n^{\backslash i} \bar{\boldsymbol{x}}_i$ and $\tau = \frac{\mu_n^2}{n} \bar{\boldsymbol{x}}_i^\top \left( \bar{\boldsymbol{S}}_n^{\backslash i} \right)^2 \bar{\boldsymbol{x}}_i$. Then, from (68), we have

$$
\sup_i \left| \frac{1}{n} \operatorname{tr} \left( \bar{\boldsymbol{S}}_n \left( \frac{1}{n} \bar{\boldsymbol{X}}_P^\top \bar{\boldsymbol{X}}_P \right) \bar{\boldsymbol{S}}_n - \bar{\boldsymbol{S}}_n^{\backslash i} \left( \frac{1}{n} \bar{\boldsymbol{X}}_P^\top \bar{\boldsymbol{X}}_P - \frac{1}{n} \bar{\boldsymbol{x}}_i \bar{\boldsymbol{x}}_i^\top \right) \bar{\boldsymbol{S}}_n^{\backslash i} \right) \right|
$$

$$
= \sup_i \left| \frac{\tau}{\mu_n^2 n} - \frac{2}{n} \cdot \left( \operatorname{tr} \left( \boldsymbol{M}_i \frac{\frac{\mu_n}{n} \bar{\boldsymbol{x}}_i \bar{\boldsymbol{x}}_i^\top \bar{\boldsymbol{S}}_n^{\backslash i}}{\mu_n + \rho} \right) + \frac{\rho}{\mu_n + \rho} \frac{\tau}{\mu_n^2} \right) + \frac{1}{n} \frac{\tau}{(\mu_n + \rho)^2} \cdot \frac{1}{n} \bar{\boldsymbol{x}}_i^\top \boldsymbol{M}_i \bar{\boldsymbol{x}}_i + \frac{1}{\mu_n^2 n} \frac{\rho^2 \tau}{(\mu_n + \rho)^2} \right|
$$

$$
= \sup_i \left| \frac{\left( (\mu_n + \rho)^2 - 2\rho(\mu_n + \rho) + \rho^2 \right) \tau}{\mu_n^2 n (\mu_n + \rho)^2} - \frac{2}{n} \cdot \operatorname{tr} \left( \boldsymbol{M}_i \frac{\frac{\mu_n}{n} \bar{\boldsymbol{x}}_i \bar{\boldsymbol{x}}_i^\top \bar{\boldsymbol{S}}_n^{\backslash i}}{\mu_n + \rho} \right) + \frac{1}{n} \frac{\tau}{(\mu_n + \rho)^2} \cdot \frac{1}{n} \bar{\boldsymbol{x}}_i^\top \boldsymbol{M}_i \bar{\boldsymbol{x}}_i \right|
$$

$$
\leq \sup_i \left( \frac{\tau}{n(\mu_n + \rho)^2} + \frac{2}{n(\mu_n + \rho)} \| \boldsymbol{M}_i \|_2 \cdot \frac{\mu_n}{n} \| \bar{\boldsymbol{x}}_i \bar{\boldsymbol{x}}_i^\top \bar{\boldsymbol{S}}_n^{\backslash i} \|_2 + \frac{\tau}{n(\mu_n + \rho)^2} \cdot \frac{1}{n} \| \bar{\boldsymbol{x}}_i \|_2^2 \| \boldsymbol{M}_i \|_2 \right)
$$

$$
\leq \sup_i \left( \frac{\tau}{n(\mu_n + \rho)^2} + \frac{2}{n(\mu_n + \rho)} \| \boldsymbol{M}_i \|_2 \cdot \sqrt{\frac{1}{n} \| \bar{\boldsymbol{x}}_i \|_2^2 \cdot \tau} + \frac{\tau}{n(\mu_n + \rho)^2} \cdot \frac{1}{n} \| \bar{\boldsymbol{x}}_i \|_2^2 \| \boldsymbol{M}_i \|_2 \right).
$$
(69)

Our next step is to bound $\rho, \tau, \sup_i \| \bar{\boldsymbol{x}}_i \|_2$ and $\sup_i \| \boldsymbol{M}_i \|_2$. Since the $\bar{\boldsymbol{x}}_i$ are standard Gaussian vectors, standard $\chi^2$ tail bounds [10] establish that $\sup_i \frac{1}{n} \| \boldsymbol{x}_i \|_2^2 = O_{\mathrm{p}}(\ln N)$. Then, by (48), we know

$$
\tau = \frac{\mu_n^2}{n} \bar{\boldsymbol{x}}_i^\top \left( \bar{\boldsymbol{S}}_n^{\backslash i} \right)^2 \bar{\boldsymbol{x}}_i \leq \mu_n^2 \cdot O_{\mathrm{p}}(\ln N) \cdot O_{\mathrm{p}} \left( \frac{1}{\mu_n^2} \right) = O_{\mathrm{p}} (\ln N).
$$

493 Using Proposition 1, Proposition 2, and (47), we also have $\rho = \Theta_{\mathrm{p}}(1)$. Finally, by (63), we have
494 $\sup_i \|M_i\| = O_{\mathrm{p}}(1)$. Plug in these results in (69), we have

$$\sup_i \left| \frac{1}{n} \operatorname{tr} \left( \bar{S}_n \left( \frac{1}{n} \bar{X}_P^\top \bar{X}_P \right) \bar{S}_n - \bar{S}_n^{\backslash i} \left( \frac{1}{n} \bar{X}_P^\top \bar{X}_P - \frac{1}{n} \bar{x}_i \bar{x}_i^\top \right) \bar{S}_n^{\backslash i} \right) \right| = O_{\mathrm{p}} \left( \frac{\ln^2 N}{N} \right).$$

495 Hence (67) holds. $\qquad \square$

## B.4   Proof of Lemma 1

497 The first part of the lemma, Equation (40), follows from Theorem 2.38 of [18].

498 For the second part, to lower bound the minimum eigenvalue $\lambda_{\min}$ of $\frac{1}{n} \bar{X} H \bar{X}^\top$, we need to find the
499 support of $\mathcal{F}$. From Section 4 of [16], we have

$$z \in \operatorname{supp}(\mathcal{F})^c \quad \Leftrightarrow \quad m(z) \in B \text{ and } \frac{1}{m(z)^2} - \gamma \int_{\eta_1}^\infty \frac{t^2 f_h(t)\,\mathrm{d}t}{(1+tm(z))^2} > 0,$$

500 where $B := \{m : m \neq 0, -m^{-1} \in \operatorname{supp}(\mathcal{H})^c\}$.

501 To show $\lambda_{\min} > c_\epsilon > 0$ holds in probability for some small enough constant $c_\epsilon$, we just need to show
502 that for all $0 \le z \le c_\epsilon$,

$$m(z) > 0 \quad \text{and} \quad \frac{1}{m(z)^2} - \gamma \int_{\eta_1}^\infty \frac{t^2}{(1+t \cdot m(z))^2} \cdot f_h(t)\,\mathrm{d}t > 0. \tag{70}$$

503 Note that the equation (40) defining $m(z)$, i.e.,

$$m(z) = -\left( z - \gamma \int_{\eta_1}^\infty \frac{t f_h(t)\,\mathrm{d}t}{1+t \cdot m(z)} \right)^{-1}, \quad \forall z \in \operatorname{supp}(\mathcal{F})^c$$

504 is equivalent to

$$z = \gamma \int_{\eta_1}^\infty \frac{t}{1+t \cdot m(z)} \cdot f_h(t)\,\mathrm{d}t - \frac{1}{m(z)}, \quad \forall z \in \operatorname{supp}(\mathcal{F})^c$$

505 Let us consider the "inverse" of $m(z)$ defined by the following equation:

$$z(m) := \gamma \int_{\eta_1}^\infty \frac{t}{1+t \cdot m} \cdot f_h(t)\,\mathrm{d}t - \frac{1}{m}.$$

506 Note that

$$\inf_{m<0} z(m) \ge \gamma > 1.$$

507 Hence, for all $z \le 1$, if $m(z)$ exists, we have $m(z) > 0$. Further, note that

$$\frac{\mathrm{d}z(m)}{\mathrm{d}m} > 0 \quad \Leftrightarrow \quad \frac{1}{m(z)^2} - \gamma \int_{\eta_1}^\infty \frac{t^2}{(1+t \cdot m(z))^2} \cdot f_h(t)\,\mathrm{d}t > 0$$

$$\Leftrightarrow \quad \gamma \int_{\eta_1}^\infty \frac{t^2}{(m^{-1}+t)^2} \cdot f_h(t)\,\mathrm{d}t < 1.$$

508 Moreover, $\gamma \int_{\eta_1}^\infty \frac{t^2}{(m^{-1}+t)^2} \cdot f_h(t)\,\mathrm{d}t$ is a continuous increasing function of $m$ with

$$\gamma \int_{\eta_1}^\infty \frac{t^2}{(m^{-1}+t)^2} \cdot f_h(t)\,\mathrm{d}t \rightarrow 0 \quad \text{as } m \to 0$$

$$\gamma \int_{\eta_1}^\infty \frac{t^2}{(m^{-1}+t)^2} \cdot f_h(t)\,\mathrm{d}t \rightarrow \gamma > 1 \quad \text{as } m \to \infty.$$

509 Therefore, we know there exists a constant $m_c$ such that for all $0 < m < m_c$, $z(m)$ is a strictly
510 increasing function on $m \in (0, m_c)$ and strictly decreasing function on $m \in [m_c, \infty)$. Thus, the
511 conditions in (70) (with $m$ in place of $m(z)$) are met for all $0 < m < m_c$. Note that

$$
m \cdot z(m) = \left( \gamma \int_{\eta_1}^{\infty} \frac{t}{1/m + t} \cdot f_h(t) \, \mathrm{d}t - 1 \right) \to \begin{cases} -1, & \text{as } m \to 0^+ \\ \gamma - 1 > 0, & \text{as } m \to +\infty \end{cases},
$$

512 Therefore, we have $z(m) \to -\infty$ as $m \to 0^+$ and $z(m) \to 0^+$ as $m \to \infty$. Then, by continuity
513 of the function $z(m)$, we know for any non-positive value $z$, the mapping between $z$ and $m > 0$
514 defined by (40) is an one to one mapping. Moreover, since the function $z(m)$ is increasing on
515 $(0, m_c)$ and decreasing on $[m_c, \infty)$, there exists an unique $m^*$ such that $z(m^*) = 0$ and $z(m)$ is a
516 continuous and increasing function on $[0, m^*]$. Hence, we have $m^* < m_c$. This implies $m(z)$ is a
517 continuous increasing function on $z \leq 0$. Further, we can find a small enough constant $\epsilon > 0$ such
518 that $m^* + \epsilon < m_c$ and $0 < z(m^*) < 1$ ($z$ is a function here). With $c_\epsilon := z(m^* + \epsilon)$, we have that
519 for all $0 \leq z \leq c_\epsilon$, the conditions in (70) are met. Hence $\lambda_{\min} > c_\epsilon > 0$ holds in probability.

520 Finally, by the dominated convergence theorem, we have

$$
\lim_{n \to \infty} m_n(z) = m(z), \text{ a.s. } \quad \text{and} \quad \lim_{n \to \infty} m_n'(z) = m'(z), \text{ a.s. } \text{ for } \quad \forall z < 0.
$$

521 For an increasing sequence $z_n \to 0^-$, note that for all $\epsilon' > 0$, we have $|m_n(z_n) - m_n(-\epsilon')| \leq \frac{\epsilon' - z_n}{c_\epsilon^2}$
522 holds in probability. Further, $m_n(-\epsilon') \to m(-\epsilon')$ almost surely and $m(-\epsilon') \to m(0)$ as $\epsilon' \to 0$.
523 Hence, for all $\epsilon' > 0$, we can choose a small enough $\epsilon'' > 0$ such that

$$
\mathbb{P}(|m_n(z_n) - m_n(-\epsilon'')| \leq \frac{\epsilon'}{3}) \to 1
$$

$$
\mathbb{P}(|m_n(-\epsilon'') - m(-\epsilon'')| \leq \frac{\epsilon'}{3}) \to 1
$$

$$
|m(-\epsilon'') - m(0)| \leq \frac{\epsilon'}{3}.
$$

524 Hence, we have $m_n(z_n) \xrightarrow{\text{P}} m(0)$. Similarly, we have $m_n'(z_n) \xrightarrow{\text{P}} m'(0)$.

## B.5 Proof of Lemma 2

526 Let $\sigma_n$ be the random variable that follows the empirical eigenvalue distribution of $N^\kappa \mathbf{\Sigma}_S$. Since
527 the minimum eigenvalue of $N^\kappa \mathbf{\Sigma}_S$ is $\frac{N^\kappa}{p_2^\kappa}$ and its maximum eigenvalue is $\frac{N^\kappa}{(p_1+1)^\kappa}$. Then for all
528 $t \in [\frac{N^\kappa}{p_2^\kappa}, \frac{N^\kappa}{(p_1+1)^\kappa}]$, we have

$$
\mathbb{P}(\sigma_n > t) = \frac{1}{|S|} \sum_{i=1+p_1}^{p_2} \mathbb{1}_{\{\frac{N^\kappa}{i^\kappa} > t\}}
$$

$$
= \frac{1}{|S|} \max \left( 0, \left\lfloor \frac{N}{t^{1/\kappa}} \right\rfloor - p_1 \right)
$$

$$
= \frac{1}{|S|} \left( \left\lfloor \frac{N}{t^{1/\kappa}} \right\rfloor - p_1 \right),
$$

529 where the last inequality is due to the fact that

$$
\left\lfloor \frac{N}{t^{1/\kappa}} \right\rfloor \geq \left\lfloor \frac{N(p_1+1)}{N} \right\rfloor = \lfloor p_1 + 1 \rfloor \geq p_1.
$$

530 Hence, as $N \to \infty$, we have

$$
\mathbb{P}(\sigma_n > t) \to \begin{cases} 1, & t \leq \frac{1}{\alpha_2^\kappa} \\ \max \left( 0, \frac{1}{\alpha_2 - \alpha_1} (\frac{1}{t^{1/\kappa}} - \alpha_1) \right), & t > \frac{1}{\alpha_2^\kappa} \end{cases}.
$$

531 Hence, the probability density function for the limiting distribution of $\sigma_n$ is indeed $f(s)$ given by
532 (42).

### B.6 Proof of Lemma 3

Without loss of generality, we assume that the diagonal elements of $\boldsymbol{\Sigma}$ are in a non-increasing order. We condition on the event where the $\frac{n}{2}$ smallest diagonal elements of $\boldsymbol{\Sigma}$ are lower-bounded by $\nu$. The minimum eigenvalue of

$$\boldsymbol{S} = \left(\frac{1}{n}\bar{\boldsymbol{X}}^\top \bar{\boldsymbol{X}} + \mu\boldsymbol{\Sigma}\right),$$

is given by

$$\sigma_{\min}(S) = \min_{\|\boldsymbol{v}\|=1} \boldsymbol{v}^\top \left(\frac{1}{n}\bar{\boldsymbol{X}}^\top \bar{\boldsymbol{X}} + \mu\boldsymbol{\Sigma}\right)\boldsymbol{v}.$$

Let $\boldsymbol{v} = (\boldsymbol{v}_1, \boldsymbol{v}_2)$ where $\boldsymbol{v}_1$ is the first $p - \frac{n}{2}$ number of components of $\boldsymbol{v}$ and $\boldsymbol{v}_2$ is the last $\frac{n}{2}$ number of components of $\boldsymbol{v}$. If $\|\boldsymbol{v}_1\|^2 \geq \frac{1}{400\gamma^2}$, then immediately, we have

$$\sigma_{\min}(S) \geq \mu\|\boldsymbol{v}_1\|^2\nu \geq \mu\frac{\nu}{400\gamma^2}.$$

Otherwise, let $\bar{\boldsymbol{X}} = (\bar{\boldsymbol{X}}_1, \bar{\boldsymbol{X}}_2)$ where $\bar{\boldsymbol{X}}_1$ is the first $p - \frac{n}{2}$ columns of $\bar{\boldsymbol{X}}$ and $\bar{\boldsymbol{X}}_2$ is the last $\frac{n}{2}$ columns of $\bar{\boldsymbol{X}}$. Then we have

$$\sigma_{\min}(S) \geq \min_{\|\boldsymbol{v}\|=1, \|\boldsymbol{v}_1\|^2 < \frac{1}{400\gamma^2}} \frac{1}{n}\|\bar{\boldsymbol{X}}_2\boldsymbol{v}_2\|^2 + \frac{1}{n}\|\bar{\boldsymbol{X}}_1\boldsymbol{v}_1\|^2 - 2\frac{1}{n}\|\bar{\boldsymbol{X}}_1\boldsymbol{v}_1\| \cdot \|\bar{\boldsymbol{X}}_2\boldsymbol{v}_2\|$$

$$= \min_{\|\boldsymbol{v}\|=1, \|\boldsymbol{v}_1\|^2 < \frac{1}{400\gamma^2}} \left(\frac{1}{\sqrt{n}}\|\bar{\boldsymbol{X}}_2\boldsymbol{v}_2\| - \frac{1}{\sqrt{n}}\|\bar{\boldsymbol{X}}_1\boldsymbol{v}_1\|\right)^2.$$

Note that $\boldsymbol{X}_2$ is a $n \times \frac{n}{2}$ standard Gaussian matrix and therefore the minimum eigenvalue of $\frac{1}{n}\boldsymbol{X}_2^\top \boldsymbol{X}_2$ can be lower bounded away from 0. Further $\boldsymbol{X}_1$ is a $n \times (p - \frac{n}{2})$ standard Gaussian matrix with $\frac{p - \frac{n}{2}}{n} \to \gamma - \frac{1}{2}$ as $p, n \to \infty$. Hence the maximum eigenvalue of $\frac{1}{n}\boldsymbol{X}_1^\top \boldsymbol{X}_1$ can be upper bounded. In fact, from Lemma 4 (Lemma 10 of [19]), we have with probability $1 - cn^2\exp(-c'n)$, we have

$$\min_{\|\boldsymbol{v}\|=1} \frac{1}{n}\|\bar{\boldsymbol{X}}_2\boldsymbol{v}\|^2 \geq \frac{1}{25} \quad \text{and} \quad \max_{\|\boldsymbol{v}\|=1} \frac{1}{n}\|\bar{\boldsymbol{X}}_1\boldsymbol{v}\|^2 \leq 9\gamma^2.$$

Hence, we have

$$\sqrt{\sigma_{\min}(S)} \geq \min_{\|\boldsymbol{v}\|=1, \|\boldsymbol{v}_1\|^2 < \frac{1}{400\gamma^2}} \frac{1}{\sqrt{n}}\|\bar{\boldsymbol{X}}_2\boldsymbol{v}_2\| - \frac{1}{\sqrt{n}}\|\bar{\boldsymbol{X}}_1\boldsymbol{v}_1\|$$

$$\geq \frac{1}{5}\sqrt{1 - \frac{1}{400\gamma^2}} - 3\gamma \cdot \frac{1}{20\gamma}$$

$$\geq \frac{\sqrt{399}}{100} - \frac{3}{20} > 0.$$

This completes the proof of this lemma.

## C  Analysis under polynomial eigenvalue decay with noise $\sigma > 0$

In this section, we consider analogues of Theorem 1–Theorem 3 that permit noisy independent observations

$$y_i = \boldsymbol{x}_i^\top \boldsymbol{\theta} + w_i, \quad i = 1, \ldots, n,$$

where $\boldsymbol{w} = (w_1, \ldots, w_n) \sim \mathcal{N}(\boldsymbol{0}, \sigma^2\boldsymbol{I})$, where we allow $\sigma^2 > 0$.

**Theorem 5.** *Assume A.1 with constant $\kappa$ and A.2 with constants $\alpha$ and $\beta$.*

*(i) We have for all $\alpha < \beta$,*

$$\mathbb{E}_{\boldsymbol{w}, \boldsymbol{\theta}}[\text{Error}] \xrightarrow{\text{p}} \left(N^{1-\kappa}\int_\alpha^1 t^{-\kappa}\,\mathrm{d}t + \sigma^2\right) \cdot \frac{\beta}{\beta - \alpha} =: \mathcal{R}_\kappa(\alpha, \sigma), \quad \forall \alpha < \beta. \tag{71}$$

When $\kappa > 1$, the minimum of $\mathcal{R}_\kappa(\alpha, \sigma)$ is achieved at $\alpha = 0$ and the minimum risk is given by

$$\min_{\alpha < \beta} \mathcal{R}_\kappa(\alpha, \sigma) = \sigma^2. \tag{72}$$

When $\kappa \leq 1$, we have nearly the same results as in Theorem 1, i.e., the minimum of $\mathcal{R}_\kappa(\alpha, \sigma)$ is achieved at $\alpha^*$ which is the unique solution of the equation $h_\kappa(\alpha) = 0$ on $(0, \beta)$, where $h_\kappa(\alpha)$ is given by

$$h_\kappa(\alpha) := \frac{\beta}{\alpha} - \int_\alpha^1 t^{\kappa-2} \, dt - 1 - \sigma^2 \mathbb{1}_{\{\kappa=1\}}. \tag{73}$$

The minimum risk is therefore given by

$$\min_{\alpha < \beta} \mathcal{R}_\kappa(\alpha, \sigma) = N^{1-\kappa} \frac{\beta}{(\alpha^*)^\kappa}. \tag{74}$$

*(ii) We have for all $\alpha > \beta$,*

$$\mathbb{E}_{\boldsymbol{w}, \boldsymbol{\theta}}[\text{Error}] \overset{\text{p}}{\to} N^{1-\kappa} \frac{\beta}{m_\kappa(0)} + \left( N^{1-\kappa} \int_\alpha^1 t^{-\kappa} \, dt + \sigma^2 \right) \frac{m_\kappa'(0)}{m_\kappa^2(0)} =: \mathcal{R}_\kappa(\alpha, \sigma). \tag{75}$$

*(iii) When $\kappa > 1$, the minimum risk for all $\alpha < 1$ and $\alpha \neq \beta$ is achieved at $\alpha = 0$, i.e., $p = o(n)$. When $\kappa < 1$, let $\alpha^*$ be the minimizer of $\mathcal{R}_\kappa(\alpha, \sigma)$ over the interval $[0, \beta)$. Then $\limsup_N \mathcal{R}_\kappa(1, \sigma) / \mathcal{R}_\kappa(\alpha^*, \sigma) < 1$.*

The proof of (i) can be easily derived from (28). The proof of (ii) can be easily derived as well from (15) and (54). For the proof of (iii), note that when $\kappa < 1$, the dominant part of the risk is the same as the noiseless case, so (iii) follows from the arguments in Theorem 3. When $\kappa > 1$, the dominant part of the risk is the noise, and therefore from (34), we have

$$\min_{\alpha > \beta} \mathcal{R}_\kappa(\alpha, \sigma) \geq \min_{\alpha > \beta} \sigma^2 \frac{\beta(1 + (s_\kappa^*)^\kappa)}{\beta + (\beta - \alpha)(s_\kappa^*)^\kappa} > \sigma^2 = \mathcal{R}_\kappa(0, \sigma).$$

Further, for $N$ large enough

$$\min_{\alpha < \beta} \mathcal{R}_\kappa(\alpha, \sigma) \to \min_{\alpha < \beta} \frac{\beta}{\beta - \alpha} \sigma^2 \geq \sigma^2 = \mathcal{R}_\kappa(0, \sigma)$$

This proves (iii) in the case $\kappa > 1$.

# D  Proof of Theorem 4

## D.1  Proof of Part (i)

Since $p < n$ holds almost surely as $N \to \infty$, by excluding an additional zero probability event $p \geq n$, we can apply the same calculation in Section 2.2 and conclude that the following equation holds under our new settings, i.e.,

$$\mathbb{E}_{\boldsymbol{w}, \boldsymbol{\theta}}[\text{Error}] \overset{\text{p}}{\to} \left( \text{tr}(\boldsymbol{\Sigma}_{P^c}) + \sigma^2 \right) \frac{\beta}{\beta - \alpha(\nu)}.$$

Hence, to show (22), we just need to characterize $\text{tr}(\boldsymbol{\Sigma}_{P^c})$. By Assumption **B.1**, we have

$$\text{tr}(\boldsymbol{\Sigma}_{P^c}) = \left( \sum_{i=1}^N \sigma_i^2 \mathbb{1}_{\{c_N \sigma_i^2 \leq \nu\}} \right) \to \frac{N}{c_N} \cdot \delta \int_{\eta_1}^\nu t f(t) \, dt.$$

Hence, we have

$$\mathbb{E}_{\boldsymbol{w}, \boldsymbol{\theta}}[\text{Error}] \overset{\text{p}}{\to} \left( \frac{N}{c_N} \cdot \delta \int_{\eta_1}^\nu t f(t) \, dt + \sigma^2 \right) \frac{\beta}{\beta - \delta \int_\nu^\infty f(t) \, dt}.$$

Hence, (22) holds. Then our next step is to find the optimal $\nu^*$ in $(\nu_b, \infty)$ when $\sigma = 0$. Define

$$g_f(\nu) := \frac{\int_{\eta_1}^{\nu} t f(t)\,\mathrm{d}t}{\beta - \delta \int_{\nu}^{\infty} f(t)\,\mathrm{d}t}.$$

To minimize $\mathbb{E}_{\boldsymbol{w},\boldsymbol{\theta}}[\text{Error}]$, we just need to minimize $g_f(\nu)$ over $\nu \in (\nu_b, \infty) \bigcap \text{supp}(f)$. To do this, we analyze the first derivative of $g_f(\nu)$. Note that

$$
\begin{aligned}
\frac{\mathrm{d}g_f(\nu)}{\mathrm{d}\nu} &= \frac{\nu f(\nu)}{\beta - \delta \int_{\nu}^{\infty} f(t)\,\mathrm{d}t} - \frac{\delta f(\nu) \int_{\eta_1}^{\nu} t f(t)\,\mathrm{d}t}{\left(\beta - \delta \int_{\nu}^{\infty} f(t)\,\mathrm{d}t\right)^2} \\
&= \frac{f(\nu)}{\left(\beta - \delta \int_{\nu}^{\infty} f(t)\,\mathrm{d}t\right)^2}\left(\nu\beta - \nu\delta \int_{\nu}^{\infty} f(t)\,\mathrm{d}t - \delta \int_{\eta_1}^{\nu} t f(t)\,\mathrm{d}t\right) \\
&= \frac{f(\nu)}{\left(\beta - \delta \int_{\nu}^{\infty} f(t)\,\mathrm{d}t\right)^2} h_f(\nu), \quad \forall \nu \in (\nu_b, \infty)\bigcap\text{supp}(f).
\end{aligned}
$$

Therefore, the sign of $\frac{\mathrm{d}g_f(\nu)}{\mathrm{d}\nu}$ is the same as the sign of $h_f(\nu)$ on $\nu \in (\nu_b, \infty)\bigcap\text{supp}(f)$. Further, note that

$$\frac{\mathrm{d}h_f(\nu)}{\mathrm{d}\nu} = \beta - \delta \int_{\nu}^{\infty} f(t)\,\mathrm{d}t > 0, \quad \forall \nu \in (\nu_b, \infty)\bigcap\text{supp}(f).$$

Hence $h_f(\nu)$ is a strictly increasing function of $\nu$ in $(\nu_b, \infty)\bigcap\text{supp}(f)$. Further, note that

$$\lim_{\nu \to \nu_b} h_f(\nu) = -\delta \int_{\eta_1}^{\nu_b} t f(t) < 0.$$

Hence, by continuity of $h_f(\nu)$, either equation $h_f(\nu) = 0$ admits an unique solution denoted by $\nu^*$ on $(\nu_b, \infty)\bigcap\text{supp}(f)$ or $h_f(\nu) < 0$ holds for all $\nu \in (\nu_b, \infty)\bigcap\text{supp}(f)$. Hence, the minimum risk is achieved at $\nu = \nu^*$ if $\nu^*$ exists. Otherwise, it is achieved at any $\nu \in \mathbb{R}\bigcup\{+\infty\}$ such that $\int_{\nu}^{\infty} f(s)\,\mathrm{d}s = 0$. Hence, if $\nu^*$ exists, the value of the minimum risk given by

$$\mathbb{E}_{\boldsymbol{w},\boldsymbol{\theta}}[\text{Error}] \xrightarrow{\text{P}} \frac{N}{c_N} \cdot \frac{\beta}{\beta - \delta \int_{\nu^*}^{\infty} f(t)\,\mathrm{d}t} \cdot \delta \int_{\eta_1}^{\nu^*} t f(t)\,\mathrm{d}s = \frac{N}{c_N} \cdot \beta\nu^*,$$

where the last equation is due to the fact that $h_f(\nu^*) = 0$. Otherwise, the value of the minimum risk given by

$$\mathbb{E}_{\boldsymbol{w},\boldsymbol{\theta}}[\text{Error}] \xrightarrow{\text{P}} \frac{N}{c_N}\delta \int_{\eta_1}^{\infty} t f(t)\,\mathrm{d}t.$$

### D.2 Proof of Part (ii)

We apply the same strategy for the proof of Theorem 2. Since the proof is similar to the proof we have shown for Theorem 2 in Section 2.3 and Appendix B, we only address a few differences here.

From Section 2.3, we should first show that equation $q_f(s, \nu) = 0$ admits an unique solution on $(0, \infty)$. Note that

$$\frac{\partial q_f(s, \nu)/s}{\partial s} = \delta \int_{\nu}^{\infty} \frac{t f(t)}{(s+t)^2}\,\mathrm{d}t > 0. \tag{76}$$

Hence, $q_f(s, \nu)/s$ is a strictly increasing function of $s$ on $s \in (0, \infty)$. Further, since $\nu < \nu_b$, we have

$$
\begin{aligned}
\lim_{s \to 0} \frac{q_f(s, \nu)}{s} &= \beta - \delta \int_{\nu}^{\infty} f(t)\,\mathrm{d}t = < 0, \\
\lim_{s \to \infty} \frac{q_f(s, \nu)}{s} &= \beta - 0 > 0. \tag{77}
\end{aligned}
$$

Hence, by continuity of function $q_f(s, \nu)/s$, we know $q_f(s, \nu)/s = 0$ admits a unique solution denoted by $s_f^*$ on $(0, \infty)$.

Note that with the same proof shown in Section 2.3, we have

$$
\mathbb{E}_{\boldsymbol{w},\boldsymbol{\theta}}[\text{Error}] = \left( \underbrace{\text{tr}\left( \boldsymbol{\Sigma}_P \left( \boldsymbol{I} - \boldsymbol{P}_{\boldsymbol{X}_P}^{\perp} \right) \right)}_{\text{part 1}} + \underbrace{\text{tr}\left( \boldsymbol{X}_{P^c}^{\top} \left( \boldsymbol{X}_P \boldsymbol{X}_P^{\top} \right)^{-1} \boldsymbol{X}_P \boldsymbol{\Sigma}_P \boldsymbol{X}_P^{\top} \left( \boldsymbol{X}_P \boldsymbol{X}_P^{\top} \right)^{-1} \boldsymbol{X}_{P^c} \right) + \text{tr}\left( \boldsymbol{\Sigma}_{P^c} \right)}_{\text{part 2}} \right)
$$

$$
+ \sigma^2 \underbrace{\left( \text{tr}\left( \left( \boldsymbol{X}_P \boldsymbol{X}_P^{\top} \right)^{-1} \boldsymbol{X}_P \boldsymbol{\Sigma}_P \boldsymbol{X}_P^{\top} \left( \boldsymbol{X}_P \boldsymbol{X}_P^{\top} \right)^{-1} \right) + 1 \right)}_{\text{part 3}}.
$$

To calculate part 1, we employ the proof strategy shown in Appendix B.2 with the following remarks. First, the expression for $\alpha$ is now given by

$$
\alpha(\nu) = \int_{\nu}^{\infty} f(t)\, \mathrm{d}t.
$$

Second, we should choose $\mu_n = \min(\frac{1}{\sqrt{N}}, o(1/c_N))$ instead of $\mu_n = \min(\frac{1}{\sqrt{N}}, o(N^{-\kappa}))$. Third, to directly apply Lemma 1, we require $\delta = 1$ from Assumption **B.1**. Yet, since we restrict $\beta < \delta$ in Assumption **B.2**, it is straightforward to extend the results in Lemma 1 to handle the case where $\delta \in (0, 1)$ by following the proof presented in Appendix B.4. The results of Lemma 2 is directly assumed by Assumption **B.1**. Finally to apply Lemma 3, we require $\frac{n}{2}$ smallest eigenvalue of $(c_N \boldsymbol{\Sigma}_P)^{-1}$ is lower bounded by a positive constant. This can be easily verified due to Assumption **B.1** and the restriction on $\beta < \delta$. Hence, follow the proof in Appendix B.2 with these remarks, we can conclude that

$$
\text{part 1} \xrightarrow{\text{P}} \frac{N}{c_N} \cdot \frac{\beta}{m_f(0)}, \tag{78}
$$

where $m_f(-\mu)$, the Stieltjes transform of the limiting spectral distribution of the matrix $\frac{1}{n} \tilde{\boldsymbol{X}} \tilde{\boldsymbol{X}}^{\top}$, is given by

$$
\mu = \frac{1}{m_f(-\mu)} - \frac{\alpha(\nu)}{\beta} \cdot \frac{\int_{\nu}^{\infty} \frac{t f(t)}{1 + t \cdot m_f(-\mu)}\, \mathrm{d}t}{\int_{\nu}^{\infty} f(t)\, \mathrm{d}t},
$$

which is equivalent to

$$
\mu = \frac{1}{m_f(-\mu)} - \frac{\delta}{\beta} \cdot \int_{\nu}^{\infty} \frac{t f(t)}{1 + t \cdot m_f(-\mu)}\, \mathrm{d}t. \tag{79}
$$

Therefore, we know $m_f^* = m_f(0) > 0$ is the solution of the following equation

$$
0 = \frac{\beta}{m_f^*} - \frac{\delta}{m_f^*} \int_{\nu}^{\infty} \frac{t f(t)}{1/m_f^* + t}\, \mathrm{d}t = q_f\left( \frac{1}{m_f^*}, \nu \right). \tag{80}
$$

Then $s = \frac{1}{m_f^*}$ should be the solution of equation $q_f(s, \nu) = 0$. By uniqueness of $s_f^*$, we have $s_f^* = \frac{1}{m_f^*}$.

For part 2 and part 3, we employ the proof strategy shown in Appendix B.3 with a few remarks. First, note that due to Assumption **B.1**, we have

$$
\text{tr}\left( c_N \boldsymbol{\Sigma}_{P^c} \right) \to N \cdot \delta \int_{\eta_1}^{\nu} t f(t)\, \mathrm{d}t \quad \text{and} \quad \text{tr}\left( c_N^2 \boldsymbol{\Sigma}_{P^c}^2 \right) \to N \cdot \delta \int_{\eta_1}^{\nu} t^2 f(t)\, \mathrm{d}t.
$$

Hence, we have the following analogue of (53):

$$
\text{part 2} \xrightarrow{\text{P}} \frac{N}{c_N} \cdot \delta \int_{\eta_1}^{\nu} t f(t)\, \mathrm{d}t \cdot (\psi + 1) + O_{\text{p}}\left( \frac{\sqrt{N}}{c_N} \cdot \psi \int_{\eta_1}^{\nu} t^2 f(t)\, \mathrm{d}t \right),
$$

where $\psi = \mathrm{tr}\left(\boldsymbol{\Sigma}_P \boldsymbol{X}_P^\top \left(\boldsymbol{X}_P \boldsymbol{X}_P^\top\right)^{-2} \boldsymbol{X}_P\right)$. Finally, to show (54), we should choose $\mu_n = \min(\frac{1}{\sqrt{N}}, o(1/c_N))$ instead of $\mu_n = \min(\frac{1}{\sqrt{N}}, o(N^{-\kappa}))$. Thus, with these remarks and modifications, we can show that

$$\text{part 2} \overset{\mathrm{P}}{\to} \frac{N}{c_N} \cdot \delta \int_{\eta_1}^{\nu} t f(t)\, \mathrm{d}t \cdot \frac{m_f'(0)}{m_f^2(0)},$$

and

$$\text{part 3} \overset{\mathrm{P}}{\to} \frac{m_f'(0)}{m_f^2(0)}.$$

Hence, our last step is to characterize $m_f'(0)$ using the chain rule. Note that from (79) and (80), we have

$$-\beta z = q_f\left(\frac{1}{m_f(z)}, \nu\right)$$

Hence, taking the derivative with respect to $z$ on both sides and with the chain rule, we have

$$-\beta \;=\; \left.\frac{\partial q_f(s, \nu)}{\partial s}\right|_{s=\frac{1}{m_f(z)}} \cdot \left(-\frac{m_f'(z)}{(m_f(z))^2}\right).$$

Hence, we have

$$\frac{m_f'(0)}{m_f^2(0)} \;=\; \left(\left.\frac{\partial q_f(s, \nu)}{\partial s}\right|_{s=s_f^*}\right)^{-1} = \beta\left(\frac{q_f(s_f^*, \nu)}{s_f^*} + s_f^* \delta \int_{\nu}^{\infty} \frac{t f(t)}{(s_f^* + t)^2}\, \mathrm{d}t\right)^{-1}$$

$$=\; \beta\left(s_f^* \delta \int_{\nu}^{\infty} \frac{t f(t)}{(s_f^* + t)^2}\, \mathrm{d}t\right)^{-1},$$

where last equation is due to the fact that $q_f(s_f^*, \nu) = 0$ and $s_f^* > 0$. Hence, we have

$$\text{part 2} \overset{\mathrm{P}}{\to} \frac{N}{c_N} \cdot \beta \frac{\int_{\eta_1}^{\nu} t f(t)\, \mathrm{d}t}{s_f^* \int_{\nu}^{\infty} \frac{t f(t)}{(s_f^* + t)^2}\, \mathrm{d}t},$$

and

$$\text{part 3} \overset{\mathrm{P}}{\to} \beta\left(s_f^* \delta \int_{\nu}^{\infty} \frac{t f(t)}{(s_f^* + t)^2}\, \mathrm{d}t\right)^{-1}.$$

This completes the proof of (ii) of the theorem.

### D.3 Proof of Part (iii)

Suppose equation $h_f(\nu) = 0$ has a solution on $(\nu_b, \infty) \bigcap \mathrm{supp}(f)$. Then by comparing the two formula in (25) and (23), we just need to show $s_f^* = \frac{1}{m_f^*} < \nu^*$. Then, from (76) and (77), we have

$$\forall\, s_0 \in (0, \infty), \text{ if } q_f(s_0) > 0, \quad \text{then } s_f^* < s_0.$$

Hence, it is sufficient to show that $q_f(\nu^*) > h_f(\nu^*) = 0$. Note that $\forall \nu \geq \eta_1$

$$h_f(\nu) - q_f(\nu) \;=\; \nu\delta \int_{\eta_1}^{\infty} \frac{t f(t)}{\nu + t}\, \mathrm{d}t - \nu\delta \int_{\nu}^{\infty} f(t)\, \mathrm{d}t - \delta \int_{\eta_1}^{\nu} t f(t)\, \mathrm{d}t$$

$$=\; \delta\nu\left(\int_{\nu}^{\infty} \frac{t f(t)}{\nu + t}\, \mathrm{d}t - \int_{\nu}^{\infty} f(t)\, \mathrm{d}t\right) + \delta\left(\int_{\eta_1}^{\nu} \frac{\nu}{\nu + t} t f(t)\, \mathrm{d}t - \int_{\eta_1}^{\nu} t f(t)\, \mathrm{d}t\right) < 0.$$

Then since $\nu^* > \nu_b > \eta_1$, we have $q_f(\nu^*) > h_f(\nu^*) = 0$.

If equation $h_f(\nu) = 0$ does not have a solution on $(\nu_b, \infty) \bigcap \mathrm{supp}(f)$, then by comparing the two formula in (25) and (24), we just need to show

$$\beta s_f^* \;=\; \frac{\beta}{m_f^*} \;<\; \delta \int_{\eta_1}^{\infty} t f(t)\, \mathrm{d}t,$$

635 which is true because, due to $q_f(s_f^*) = 0$, we have

$$\beta s_f^* \quad = \quad s_f^* \delta \int_{\eta_1}^{\infty} \frac{t f(t)}{s_f^* + t} \, \mathrm{d}t \quad < \quad \delta \int_{\eta_1}^{\infty} t f(t) \, \mathrm{d}t.$$

636 Putting everything together completes the proof of part (iii).