[Reviews · NeurIPS 2019]

Reviewer 1



In the past year, a number of papers have given theoretical insight into "double descent" risk, where the minimum risk is (counterintuitively) found at some $p > n$. This paper provides "double descent" results under slightly different settings from the preceding literature. Originality: The idea of double descent is not new, but the authors explore this behavior in a slightly different setting compared to preceding literature. One useful advantage of this paper is that there is no minimal eigenvalue assumption. Quality: Claims are supported by theoretical analysis and complete. Simulations to illustrate the expected behavior would be welcome. Clarity: The paper is properly motivated, well-written, and placed within the surrounding literature. Significance: The paper is an interesting addition to the interpolation / double descent literature, but aside from analyzing its behavior under different assumptions does not pose many new ideas. *** I thank the authors for their careful reading of our reviews and for addressing my concerns.

Reviewer 2



Authors analyze the behavior of principal component regression estimates when p>n. They provide theory on the asymptotic value of the L2 error with the sample size n growing to infinity and fixed ratios between the number of features and n and the number of PCs and n. While the work is interesting and focuses on the unintuitive phenomenon, in the current form it's practical to use is not obvious. Results obtained under the assumption of polynomial decay of eigenvalues are solid. However, asymptotics there is not that interesting since at some point, the number of required eigenvectors is clearly much smaller than the number of features and the problem becomes easy even if p>n. From my understanding, the decay of eigenvalues and the growth of p is the key component making the theory hold true. It would be interesting to see where the intuition breaks or what happens when new features are equally important (i.e. have equally large eigenvalues) -- what if all eigenvalues are separated from zero. Despite a catchy title suggesting applied work, in the current version, authors do not answer explicitly the question they ask in the title. While clearly, the paper discusses the choice of p, under a set of assumptions introduced in theorems there is no clear guideline on the choice of p. The paper is well-written and easy to read. My two main concerns are: - it seems that it solves a very particular case of a much bigger problem. - it's not actionable despite a suggestive title.

Reviewer 3



In the paper, the authors discussed PCR, a well-know variant of regression models, and showed the existence of a "double descent" phenomenon. The paper is technically sound and relatively well-written. I check most of the math and they are correct and reasonable to follow. I do have some concern that too much of the space is taken by the algebra which could make it difficult for readers to grasp the high-level intuition, specifically if they do not have enough time to plough through the equations. Considering the space limit for a NeurIPS submission, I think it's better to reorganize some of the proofs to the appendix, and add a discussion/conclusion session to highlight more about the intuitions. Back to the content of the paper: the PCR model dealt in this model is different from traditional regression analysis, in that - It is high-dimensional, with n,p,N all going to infinity - It mainly concerns the generalization (i.e. expected risk, or out-of-sample performance) rather than in-sample performance (e.g. how well the model fits existing data). Now I think both differences are worth highlighting (which are actually direct causes of the "double descent" phenomenon). And these two differences extend the traditional analysis into a relevant and modern regime, especially in our current era (high-dimensional data, over-parametrized neural nets, etc.). Although linear models are simpler, understanding of their behavior in this regime could possibly shed light on the bigger questions (why over-parametrized nnets generalizes well?) I do want to discuss a bit of the high-level intuitions here, especially how the "double descent" actually arises. a) p\infty so one would pick p=N here, and the expected risk goes to 0 as n->\infty as we will eventually learn the right model. So it was actually surprising to see that the generalization risk increases as α->β which actually seems to diverge (Figure 1). Which also means the two key characteristics (high-dimensional, out-of-sample) MUST be playing important roles here. After a careful examination one would conclude that the high-dimensionality is killing the generalization risk when α->β. I believe the following point is crucial: The design matrix X, when α is close to β, is going to be ill-conditioned with high probability (think of marcenko-pastur law of its spectrum). When α=β, it is asymptotically guaranteed (even in the almost sure sense) that X will have singular values arbitrarily close to 0 as n->\infty. This essentially makes estimation very hard. b) p>n, or the second descent: This observation naturally leads to the second descent as p increases pass n. p>n actually serves as an "implicit ridge regularization" here, since it tries to look for the solution with small l2 norm. This can also be seen from the above point: as p moves away from n, X is becoming better-conditioned again. This "over-parametrization serves as implicit regularization" is a pretty interesting phenomenon, as it introduces bias to the problem (which is essentially non-exist when p

[Author Response · NeurIPS 2019]

We are grateful to the reviewers for their valuable feedback. They have clearly understood the paper well; we agree with the assessment of both the strengths and weaknesses of the paper. In our revision, we will try our best to address the concerns and execute the suggestions.

We first address the main concerns that shared by all reviewers.

- *When does double descent happen in regression?*
  Double descent happens if the risk curve has a spike at $p = n$, typically caused by a near-zero minimum eigenvalue of $X_P^\top X_P$. For example, when $X$ is an i.i.d. Gaussian matrix, and the noise is i.i.d. noise, then there will be a spike at $p = n$ and double descent happens, regardless of whether $X_P$ contains the true features or not.

- *When the second descent is better than the first one?*
  In our current setting, our theorems show that the answer depends on the noise level and the decay rate of the eigenvalues of the covariance matrix of $X$. If the eigenvalues are all equal, the second descent achieves a lower risk regardless of noise (we assume the variance of the noise does not scale with $n$). If the eigenvalues decay too fast, i.e., $\kappa > 1$, the second descent achieves a lower risk only if there is no noise.
  In general, we believe that if the noise level is high enough such that the noise becomes the dominant part of the risk or the eigenvalues of covariance matrix decay too fast, then $p < n$ regime is better. Otherwise, the $p > n$ regime may yield a lower risk.

**To Reviewer 1**

- We plan to discuss the results in the context of surrounding literature around interpolation and ridgeless regression (including future directions).
- We plan to include more discussion of simulations results.
- We plan to add more appropriate figure titles.

**To Reviewer 2**

- We plan to include more discussion of simulation results.
- We plan to discuss to what extent we can answer the question in the title, specifically for our current setting, and also what we hope to answer in future work.

**To Reviewer 3**

- We agree that our setting is non-standard for PCR, although it is plausibly realizable in a semi-supervised learning setting. (We apologize for forgetting to emphasize this important point!) Unfortunately, we don't have an answer for what happens in the standard PCR setting.
- We plan to discuss to what extent we can answer the question in the title, specifically for our current setting, and also what we hope to answer in future work.

[Meta-Review · NeurIPS 2019]

The authors study the "double-descent" phenomenon in high-dimensional principal components regression. The results are gnerally interesting and the reviews mostly positive. My own inclination is positive as well, but I have several nontrivial concerns. The authors should be careful to address these in a camera-ready version of the paper. - Title: the title is misleading. There is no practical tool being offered here for PCR, just an interesting analysis of its performance. So the title should be changed. - Presentation: several pages of the paper are taken up by lengthy proof sketches that are essentially only interesting/useful to those fluent in random matrix theory. This is a definite waste of space because most readers will not get anything from this. It would be much better to use the space to explain the **significance** of the results, the consequences for practical/general consumptions, the intuition, etc. And of course, experiments. - Experiments: are totally lacking from the main paper. There should be enough space for at least a few convincing experiments once the proof sketches are cut. - Comparison to existing work: it's not clear at all to me the significance of the lower bound on the eigenvalues of \Sigma in previous papers on ridge regression/double-descent. As far as I understand, this is typically used to invoke some kind of uniform convergence argument that allows us to interchange limits (and effectively take \lambda \to 0 before taking n,p \to \infty). But this is a rather precise/technical use of such a condition, not a fundamental reliance on a particular regime. And of course it is entirely possible to just keep the \lambda \to 0 "on the outside" and still interpret the results as making a statement in the proper order. So I feel the authors need to much better motivate what is new about their analysis if they want to claim that assumption of Gaussianity + new techniques actually makes a difference. Finally, I would like to see a more explicit comparison to the misspecified model in Hastie et al, which seems very similar.